# Combating Inter-Task Confusion and Catastrophic Forgetting by Metric Learning and Re-Using a Past Trained Model

**Sayedmoslem Shokrolahi**                                    *20ss184@queensu.ca*
*Department of Electrical and Computer Engineering*
*Queen's University*

**Il-Min Kim**                                              *ilmin.kim@queensu.ca*
*Department of Electrical and Computer Engineering*
*Queen's University*

**Reviewed on OpenReview:** *https://openreview.net/forum?id=jRbKsQ3sYO*

## Abstract

Despite the vast research on class-incremental learning (IL), the critical issues have not yet been fully addressed. In this paper, utilizing metric learning, we tackle two fundamental issues of class-incremental learning (class-IL), inter-task confusion and catastrophic forgetting, which have not been fully addressed yet in the literature. To mitigate the inter-task confusion, we propose an innovative loss by utilizing the centroids of previously learned classes as negatives and current data samples as positives in the embedding space, which reduces overlaps between the classes of the current and past tasks in the embedding space. To combat catastrophic forgetting, we also propose that the past trained model is stored and re-used for generating past data samples for only one previous task. Based on this, we further propose a novel knowledge distillation approach utilizing inter-class embedding clusters, intra-class embedding clusters, and mean square embedding distances. Extensive experiments performed on MNIST, CIFAR-10, CIFAR-100, Mini-ImageNet, and TinyImageNet show that our proposed exemplar-free metric class-IL method achieves the state-of-the-art performance, beating all baseline methods by notable margins. We release our codes as the supplementary materials.

## 1 Introduction

Incremental learning (IL) is the learning paradigm in which the model learns from sequential input data without accessing all past data. In any IL, a fundamental goal is to remember all seen experiences as much as possible at each point in time. In practice, however, IL easily suffers from a critical issue of catastrophic forgetting (CF) French (1999); De Lange et al. (2021); Zhou et al. (2024) that model entirely or substantially forgets what it has already learned. This poses a significant challenge in IL scenarios, as the model needs to adapt to new information without undermining its previously acquired knowledge.

IL settings can be largely categorized into (i) task-based and (ii) task-free van de Ven et al. (2021). The task-based approach itself includes task-IL, domain-IL, and class-IL van de Ven et al. (2021). In task-IL problems, the task identity is always available in training and test, making it the easiest setting in IL. In both domain-IL and class-IL settings, task identity is not provided at the test time. Particularly, in the class-IL, the model must infer the task identity during the test phase, which is not a requirement of the domain-IL. Meanwhile, the task identities are inaccessible during the training and inference phase in task-free cases.

In class-IL, which we focus on in this paper, not all performance degradation can be attributed solely to the phenomenon of catastrophic forgetting. Another significant factor contributing to performance decline

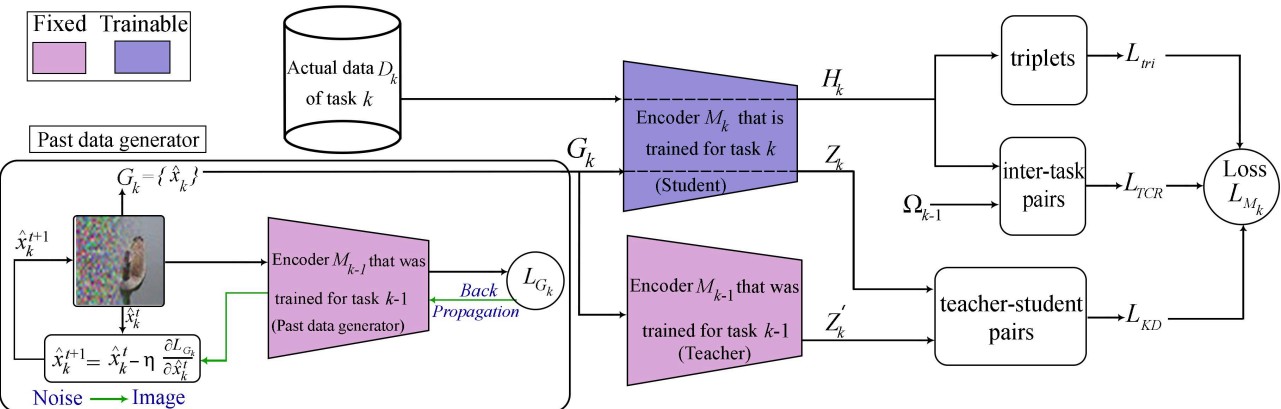

Figure 1: Training of the proposed method. To learn the current task without over-fitting, actual current data $D_k$ are supplied to the trainable encoder $M_k$ to construct triplet loss, $L_{tri}$. To avoid inter-task confusion, using the stored centroids of the past classes, $\Omega_{k-1}$, as negative samples in conjunction with positive samples extracted from the current dataset $D_k$, we construct the inter-task confusion regularizer, $L_{TCR}$. To tackle the catastrophic forgetting, we design KD loss $L_{KD}$ by supplying the synthetic past data $G_k$ to both $M_k$ and the stored (fixed) past encoder $M_{k-1}$ three main losses. In the test phase, the trained encoder $M_k$ is extracted and used as the nearest mean classifier (NCM).

is the confusion between the current task and past tasks, called the inter-task confusion (ITC) Masana et al. (2022); Huang et al. (2022; 2023); Nori & KIM (2024). Inter-task confusion arises in class-incremental learning (class-IL) when a model struggles to differentiate between classes belonging to different tasks during testing. This occurs because the classes from distinct tasks are never seen together during training, and the model is not explicitly taught to distinguish between them. As a result, the learned features are not optimized to discriminate across tasks, leading to confusion when all tasks are evaluated together without access to task-specific identifiers. For instance, while each task's classes are trained separately to distinguish their own categories, the discrepancy between classes from different tasks remains unresolved, resulting in misclassifications across tasks. This phenomenon becomes particularly evident when task-IDs must be inferred at test time.

Although these two concepts often have not been clearly distinguished in the literature, the ITC and the CF should be treated separately because the former refers to the confusion between the current and past tasks, whereas the latter refers to forgetting the past tasks. Further analysis of the impact of these two concepts on performance is presented in Subsection 4.5.

In recent works on IL, many researchers have proposed various approaches to improve the performance of class incremental learning by tackling CF. Mitigating CF can implicitly reduce ITC with different degrees, e.g., a replay base method with infinite memory should be able to address both CF and ITC. However, we argue that tackling ITC explicitly will potentially improve the performance further in realistic scenarios. Representative methods so far could be categorized into four main groups De Lange et al. (2021): (i) replay methods, (ii) regularization-based methods, (iii) parameter isolation methods, and (iv) hybrid methods. In the replay methods, part of the data from previous tasks are stored or past data are synthesized to alleviate catastrophic forgetting Rebuffi et al. (2017); Chaudhry et al. (2019). In the regularization-based methods, regularization terms are added to the loss function, possibly combined with the approach of knowledge distillation (KD) Rannen et al. (2017). In the parameter isolation methods, the parameters of the model are controlled to address the issue of forgetting by preventing interference between the current task and past tasks Serra et al. (2018).

Our proposed approach, shown in Fig. 1, can be categorized as the hybrid method combining regularization and generative replay in the class-IL setting. In our scheme, we aim to (i) mitigate the inter-task confusion

and (ii) combat catastrophic forgetting, while we use the triplet loss $L_{tri}$ to train the model $M_k$ for the current task using the current actual data $D_k$, only which we are assumed to directly access.

First, we pay a special attention to the confusion between the classes of different tasks, i.e., the inter-task confusion Masana et al. (2022), which results in significant performance degradation in the class-IL. To overcome this challenge, we propose a novel regularization term, $L_{TCR}$, using the embedding centroids of previously learned classes as negative samples and the embeddings of the data samples of the current task as positive samples. The current model $M_k$ is trained such that current classes are located away from previous classes in the embedding space using the positive and negative samples.

Second, catastrophic forgetting is a critical issue in any IL including class-IL. To mitigate the catastrophic forgetting, we store the past model $M_{k-1}$ trained in the last task time $k-1$ and use it (while freezing it) to train the current model $M_k$ in two ways. We first use $M_{k-1}$ to synthesize past data samples belong to the very last task $G_k$, which will be used for training $M_k$. Note that we do not store any actual samples of previous tasks; instead, we synthesize past data samples by storing and re-using a past trained (now, fixed) model. We also use $M_{k-1}$ as the teacher to teach the current model $M_k$, the student. This is based on the inspiration that, from the perspective of the past knowledge, $M_{k-1}$ is the model having such knowledge whereas $M_k$ is the model to be trained to learn such knowledge. For efficient knowledge distillation (KD) of the past task from $M_{k-1}$ to $M_k$, we will design a new loss, $L_{KD}$, by fully exploiting the properties of embedding structures in class-IL. The main contributions of our work can be summarized as follows:

- Using the centroids of previously learned classes as negative samples, we design a regularization term customized to overcome inter-task confusion.

- We propose a highly effective KD method in the embedding space to combat catastrophic forgetting. To this end, we also propose a method for synthesizing past data samples by storing and re-using a past trained model $M_{k-1}$. Using $M_{k-1}$, at current task $k$, we only generate samples belong to $(k-1)$-th task.

- Extensive simulations are performed, which demonstrate that our proposed scheme achieves the state-of-the-art performance in the setting of class-IL.

## 2 Related works

### 2.1 Regularization, Replay, and KD in IL

In IL, we must prevent over-fitting in each new task and, at the same time, we must stop forgetting past tasks. For the purpose of reducing forgetting in IL, general regularization methods used for handling over-fittings such as dropout Goodfellow et al. (2013) and early stopping Maltoni & Lomonaco (2019) might be used; but they are not very effective in IL. Instead, searching for important weights to keep them unchanged with regularization terms is a more effective way Bühlmann & Van De Geer (2011).

With advances in generative models, generative replay methods grab enormous attention in IL scenarios. A parallel generator and a solver model were used in Shin et al. (2017) to create synthetic images as replay data in the IL case. More complex issues were investigated in the brain-inspired method van de Ven et al. (2020). They used generated representations in the replay phase instead of using synthetic images. Cost-Free IL PourKeshavarzi et al. (2022) proposed a memory recovery method that helped the current network remember past information without storing data. Error sensitivity modulation experience replay (ESMER) Sarfraz et al. (2023), proposes that the model should prioritize learning from smaller losses to minimize significant feature drift, adjusting learning rates dynamically based on error consistency.

Transferring learned knowledge from the already trained neural network to a new raw network is proved to be useful in many machine learning applications. Most of the solutions for this kind of knowledge transfer are based on the concept of KD Hinton et al. (2015). In IL, the distillation of knowledge could be performed across tasks, and various methods used KD-based transfer learning Schwarz et al. (2018); Dhar et al. (2019); Bhat et al. (2024); Chen et al. (2024); Liang & Li (2024).

## 2.2 Soft-Max Classifiers and Metric Learning in IL

The goal of deep metric learning is to train a differentiable model $f_\theta(\cdot) : \mathcal{X} \to \mathbb{R}^p$ that maps input domain $\mathcal{X}$ to a (compressed) embedding domain $\mathbb{R}^p$ together with a distance metric $d \in \mathbb{R}$ such a way that similar data samples result in a small distance and dissimilar data samples produce a large distance. To this end, loss functions in metric learning need to be properly designed to find similarities/dissimilarities between samples in the embedding space.

In most static learning scenarios (i.e., the standard non-IL scenarios), using softmax loss in classification problems has proved to provide firm performance. When it comes to IL, however, there exists an inherent deficiency in softmax loss, because softmax loss is essentially focused on drawing lines (i.e., decision boundaries) between different classes in the embedding space. Fig. 2(a) shows the embeddings of two different classes in task 1. In task 1, because the actual data samples of the two classes are available for training, we can draw a line between the two classes, which is denoted by decision boundary 1. In the next task time (task 2), which is shown in Fig. 2(b), new actual data samples for two new classes are provided for training; so, it is again possible to draw a line between the two new classes, denoted by decision boundary 2. However, because the past data samples of task 1 are not accessible anymore, it is difficult to draw the optimal line, denoted by the ideal decision boundary, that distinguishes all the four classes seen so far. Such optimal boundary *could* be drawn only if all the data samples including the past classes *were* accessible. This is a fundamental limitation of the approach based on softmax classifiers. We note that softmax loss does not directly (or explicitly) try to minimize the size of each cluster of embeddings nor maximize the distance between clusters of embeddings in the embedding space. The fundamental difference between static learning and IL is that, in IL, we do not know where the new embedding clusters will be placed in the embedding space in the future tasks. For this reason, softmax loss might not be the best approach in IL.

In IL, an alternative approach is to use deep metric learning with a similarity/dissimilarity loss function such as a triplet loss. Minimizing the triplet loss encourages putting the same class data samples closer to each other and putting different class data samples farther from one another, in the embedding space, which results in a smaller size of each embedding cluster of each class and a larger distance between embedding clusters of different classes. Fig. 3(a) shows two embedding clusters of two classes in task 1. In task 1, by optimizing the triplet loss, the size of each cluster is minimized and the distance between the two clusters is maximized. As shown in Fig. 3(b), in task 2, two new embedding clusters of two new classes are introduced to the embedding space. Again, by using the triplet loss on the data samples of the two new classes (but not on the data samples of the past two classes), the size of each of the two new clusters is minimized and the distance between the two new clusters is maximized. Then, the chances that any of the two new clusters happen to overlap with any of the old clusters become small in the metric learning, because each of the four clusters was/is minimized, and two clusters in each task were/are separated as much as possible. This is the reason why the approach of metric learning is inherently more suitable for IL.

The triplet loss uses three instances: (i) an anchor $(x_a)$, (ii) a positive data sample $(x_p)$ having the same label as the anchor, and (iii) a negative data sample $(x_n)$ having a different label. Considering a distance metric $d(i,j)$ on the embedding space (e.g., squared Euclidean distance), the triplet loss is defined as $L_{tri}(x) = \max(0, d(x_a, x_p) - d(x_a, x_n) + m)$ for a single triplet input $(x_a, x_p, x_n)$. The goal is to satisfy $d(x_a, x_p) - d(x_a, x_n) + m < 0$, which means making $d(x_a, x_p)$ smaller than $d(x_a, x_n)$ by a predefined margin $m$. During inference, we use the nearest class mean (NCM) classifier Yu et al. (2020).

## 3 Proposed Scheme

### 3.1 Incremental Learning Formulation

At the initial stage, a model $M_0$ is trained from scratch in the standard *non-incremental* manner using training dataset $D_0$ containing $C_0$ classes in a classification application. Then new datasets $D_k$, each having additional new $C$ classes, are sequentially collected and supplied to the model $M_k$ one by one so that $M_k$ is trained only on $D_k$ at each task in the *incremental* manner for $k = 1, \ldots, K$, where $K$ is the total number of *incremental tasks*. The current model $M_k$ does not have any access to the actual past data, $D_i$, $i = 0, 1, \ldots, k - 1$.

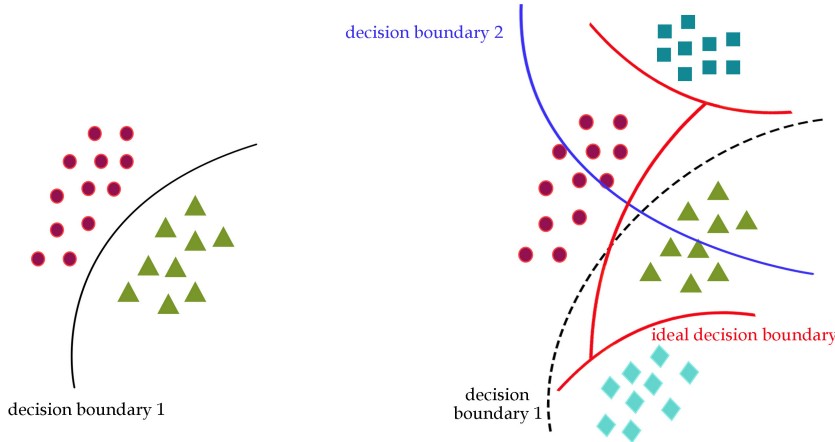

(a) Embeddings of two classes in task 1.

(b) Extra embeddings of two additional classes are added in task 2.

Figure 2: Softmax classifiers.

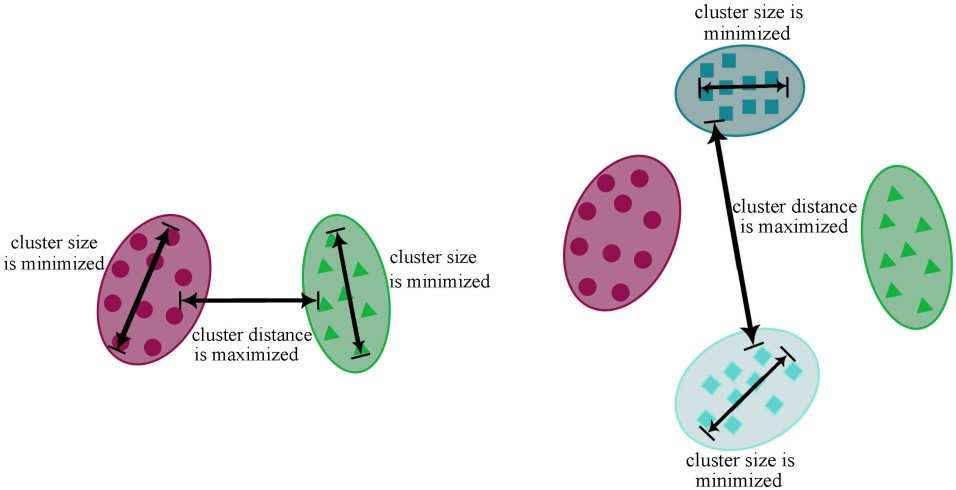

(a) Embeddings of two classes in task 1.

(b) Extra embeddings of two additional classes are added in task 2.

Figure 3: Metric learning approach.

For performance evaluation in the test phase, we use the average accuracy and forgetting, which are the widely adopted standard performance metric in IL Chaudhry et al. (2019; 2018a); Yu et al. (2020). After training all $K$ tasks, the average accuracy $A_K$ is given by $A_K = \frac{1}{K} \sum_{j=1}^{K} a_{K,j}$, where $a_{i,j}$ is the test accuracy of the model that has been incrementally trained from task 1 to $i$ on the held-out test set of task $j$.

Forgetting is defined for the $j$-th task after the model has been trained incrementally up to task $k$, where $k > j$, as in Chaudhry et al. (2018a): $o_j^k = \max_{l \in \{1,\dots,k-1\}} (a_{l,j} - a_{k,j}), \quad \forall j < k$. Considering $o_j^k \in [-1, 1]$, defined for $j < k$, as our focus lies on quantifying forgetting across preceding tasks, and $a_{k,j}$ is the test

accuracy of the model that has been incrementally trained on the held-out test set of task $j$. Furthermore, by standardizing against the count of previously encountered tasks, the average forgetting at the $k$-th task is denoted as $O_k = \frac{1}{k-1} \sum_{j=1}^{k-1} o_j^k$.

## 3.2 Overview of Our Proposed Method

The fundamental idea of our proposed method is to intelligently control the sizes and positions of embeddings of the classes of the current task considering the embedding locations of the past tasks in the embedding space to (i) avoid inter-task confusion and (ii) prevent catastrophic forgetting. The overall structure is illustrated in Fig. 1. Our proposed model is composed of two encoders ($M_k$ and $M_{k-1}$), the current input dataset ($D_k$), the set of stored *centroids* of the past classes ($\Omega_{k-1}$), and the generated dataset ($G_k$) of the past data samples. At the current task $k$, the encoder denoted by $M_k$ is trained by using $D_k$, $\Omega_{k-1}$, and $G_k$, whereas the other encoder indicated by $M_{k-1}$, which was trained in the last task $k-1$ and has been stored, is fixed. Note that, in $\Omega_{k-1}$, only a single centroid (not multiple samples) for each past class is stored, which must be stored for NCM. In our proposed method, the current model $M_k$ is trained by $L_{tri}$ to learn the current task using the embeddings $H_k$ which are produced when the dataset $D_k$ of the actual data samples for the current task are supplied as input. To be more precise, $H_k = [h_k^1, \ldots, h_k^b] \in \mathbb{R}^{p \times b}$ denotes the normalized embedding matrix produced by trainable student model $M_k$ using real current task dataset. The dimension of embedding space is $p$ and batch size is $b$.

To address the intertask confusion, we introduce $L_{TCR}$, through which $M_k$ learns how to avoid any confusion between the current task and the past tasks utilizing previously stored embedding centroids of past classes ($\Omega_{k-1}$) as negatives in conjunction with $H_k$ as positives.

The other critical issue is to combat catastrophic forgetting. To this end, we use a data generator that is composed of input $\hat{x}$ (this is initialized as random noises) and the previously trained (now, fixed) encoder $M_{k-1}$ in order to generate the synthetic images $G_k$ of the classes of the past task(s). These synthesized past images in $G_k$ are used by $M_{k-1}$ and $M_k$ such that $M_{k-1}$ (as the teacher having strong knowledge of $G_k$) teaches $M_k$ (as the student who has not learned $G_k$ yet) about the past knowledge $G_k$. Specifically, $G_k$ is supplied as the inputs to both encoders, $M_k$ and $M_{k-1}$, to produce the embeddings $Z_k$ and $Z_k'$, respectively. These embeddings are used together to construct the KD loss, $L_{KD}$. Note that the (fixed) encoder $M_{k-1}$ is used for two different purposes in the current task: (i) to generate the synthetic images $G_k$ of the past classes and (ii) to teach $M_k$ the past knowledge as a teacher. Overall, the proposed total loss function for training $M_k$ is composed of three main terms:

$$L_{M_k} = L_{tri} + \lambda_1 L_{TCR} + L_{KD}, \tag{1}$$

where $L_{tri}$ is the triplet loss used to train the current model $M_k$ with the current available data $D_k$ (for implementation details, see Appendix), and the other two losses, $L_{TCR}$ and $L_{KD}$, are explained in more detail in what follows.

## 3.3 Inter-task Confusion Regularizer, $L_{TCR}$, to Combat Inter-task Confusion

This loss, $L_{TCR}$, is proposed to avoid any confusion between the current task and any past tasks. Ideally, class-IL must be able to accurately differentiate between classes across tasks as well as between classes within each task. In class-IL, however, it is very challenging for the model to effectively discriminate between the classes across tasks because the model is distinctively trained for each individual task. This fundamental challenge is inevitable because the model cannot see the data for the classes together in training when they belong to different tasks. This challenge in class-IL, known as inter-task confusion Masana et al. (2022), is illustrated in Fig. 4(b). Note that the triplet loss focuses solely on each task one by one, meaning that it does not make any effort to avoid collision between the cluster of a class belonging to the current task and that of another class belonging to any of the past tasks, which results in inter-task confusion. To mitigate this inter-task confusion, we propose a regularization term that is specifically designed for the class-IL paradigm.

The fundamental idea of our proposed scheme is that, in metric learning, the cluster centroids are anyway stored in the course of training to be leveraged later for inference and this information can be effectively used

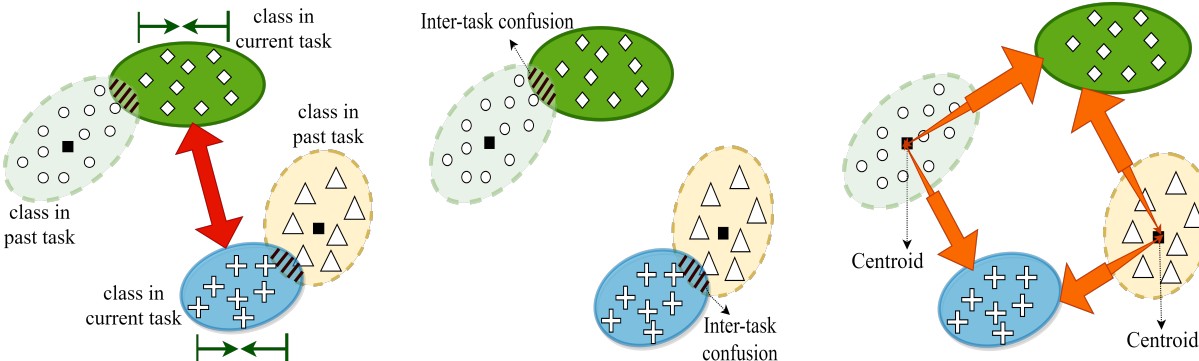

(a) In the current task, triplet loss decreases (the green arrows) the cluster size of each class and increases (the red arrow) the inter-cluster distances between any two classes of the current task in the embedding space.

(b) The clusters of classes of the current task and those of past tasks might overlap in the embedding space, which is inter-task confusion. This cannot be mitigated by triplet loss.

(c) Inter-task Confusion Regularizer, $L_{TCR}$, increases the inter-cluster distance between classes of different tasks in the embedding space. Specifically, the distances between the data samples (the hollow diamonds and crosses) of the classes of the current task and the centroids (the black squares) of the classes of all past tasks are increased (the orange arrows) by $L_{TCR}$ in the embedding space.

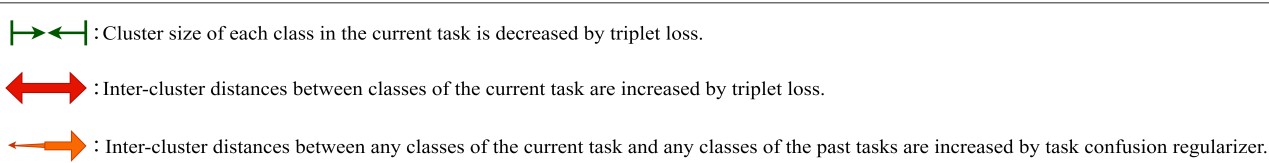

Figure 4: Triplet loss, inter-task confusion phenomenon, and inter-task confusion regularizer.

to mitigate inter-task confusion. Specifically, we propose a new loss, $L_{TCR}$, which utilizes cluster centroids of the classes of all past tasks as negative samples while the positive samples are the randomly selected data points with noise in every class of the current task. In other words, the positives are a random subset of embeddings $H_k$ which are produced when the dataset $D_k$ of the actual data samples for the current task are supplied as input to trainable model $M_k$. This loss encourages the distance between the data samples of the current classes and the centroids of the past classes in the past tasks to be larger than a predefined margin $m_{TCR}$ in order to effectively avoid the overlap between classes across distinct tasks, thereby addressing inter-task confusion, as shown in Fig. 4(c). Mathematically, the loss is given by

$$L_{TCR} = \max\left(0, m_{TCR} - d(x_p, \mu_{v'})\right), \quad \forall v' \in \Omega_{k-1}, \tag{2}$$

where $x_p$ denotes the positive samples. Also, $\mu_{v'}$ indicates the negative samples, which are the centroids of the class $v'$, and $\Omega_{k-1}$ is the set of class indices from the first task up to task $k-1$. To facilitate increasing the distance effectively, as in the triplet loss, we use a margin $m_{TCR}$. The size of a mini-batch for constructing $L_{TCR}$ is the same as in the triplet loss, which is $b$.

### 3.4 KD loss, $L_{KD}$, to Combat Catastrophic Forgetting

This loss, $L_{KD}$, is proposed to address the issue of catastrophic forgetting. It is composed of three innovative knowledge distillation loss terms for transferring the knowledge of the previous tasks that had been learned by $M_{k-1}$ (the teacher) to the current model $M_k$ (the student) as follows:

$$L_{KD} = \lambda_2 L_{KD}^{mse} + \lambda_3(L_{KD}^{intra} + L_{KD}^{inter}), \tag{3}$$

where $\lambda_2$ and $\lambda_3$ are hyper-parameters to control the impact of each term. To construct these three KD losses, we utilize the synthetic data $G_k$ of the last task as inputs to both $M_k$ and $M_{k-1}$. Therefore, we will begin by detailing the process of generating $G_k$ in the following subsection. Subsequently, each of the three KD losses will be explained one by one.

### 3.4.1 Generating Past Synthetic Data $G_k$

In IL, one of the most effective approaches to prevent catastrophic forgetting is to access some of the previous data in training. This approach, called the replay strategy, is classified into two types: (i) coreset replay Rebuffi et al. (2017) and (ii) generative replay Sun et al. (2021). In the approach of coreset replay, by storing small amount of actual previous data samples, IL is facilitated. However, this approach cannot be adopted when data privacy matters. In this case, the other approach of generative replay might be considered. GANs are a popular model to be used as generative models in IL Lesort et al. (2019); Cong et al. (2020). Variational autoencoders (VAEs) are also valuable as a generative model in both supervised and unsupervised IL Caselles-Dupré et al. (2021); Kemker & Kanan (2017). In the setting of class-IL, however, incrementally training generative models such as GANs or VAEs can be challenging because the training data is presented incrementally. Indeed, as discussed in van de Ven et al. (2020), in the class-IL setting, generative replay works well for small data sets (e.g., MNIST LeCun et al. (1998)), but scaling it up to non-small ones (e.g., CIFAR-100 Krizhevsky (2009), TinyImageNet Le & Yang (2015)) is not straightforward. Also, because such incremental training of generative models is typically distinct from incremental training of the discriminative model performing classification, the overall training complexity goes up. To address these issues, in this paper, we propose a more scalable and effective approach for generative replay without additionally training a dedicated separate generative model.

The core idea of the proposed replay approach is to store and re-use a previously trained model for generating synthetic past images. Specifically, the model $M_{k-1}$ that was trained in the last task time $k-1$ is stored and used for the purpose of generating past data samples, which in turn are used to help the training of the current model $M_k$, as shown in Fig. 1. Note that we do not introduce any separate or distinct generator such as GANs or VAEs for generating past data samples; instead, at each task time, we store *a single* (past trained) model and *re-use* it for the training of the next model. Here, the critical question is how to generate past data samples using a trained discriminator $M_{k-1}$, which is discussed in the following.

For generating class-conditional synthetic images using a (fixed) trained model, in Yin et al. (2020), a very effective method, called the Deep-Inversion, was proposed. In this paper, we also follow the approach of Yin et al. (2020) for generating class-conditional synthetic images. The fundamental idea of Yin et al. (2020) is to minimize the distances of the means and the distances of the variances between the actual images $x \in D$ and the synthesized (generated) images $\hat{x} \in G$. In training, $\hat{x}$ is initialized by random (Gaussian) noise and gradually updated to get close to the actual image, $x$. Mathematically, for a model that was trained on $D$, the following regularization term $R_b(\hat{x})$ is used Yin et al. (2020):

$$R_b(\hat{x}) = \sum_l \parallel \psi_l(\hat{x}) - \mathbb{E}(\psi_l(x)|D) \parallel + \sum_l \parallel \sigma_l^2(\hat{x}) - \mathbb{E}(\sigma_l^2(x)|D) \parallel, \tag{4}$$

where $\psi_l(\hat{x})$ and $\sigma_l^2(\hat{x})$ are the mini-batch-wise mean and variance, respectively, at the $l^{th}$ convolutional layer when the input to the (fixed) model is $\hat{x}$. In the same way, $\psi_l(x)$ and $\sigma_l^2(x)$ are defined for $x$. In the setting of IL, however, it is not possible to exactly determine $\psi_l(x)$ and $\sigma_l^2(x)$, because the model does not have any access to the original past dataset $D$. Nevertheless, $\mathbb{E}(\psi_l(x)|D)$ and $\mathbb{E}(\sigma_l^2(x)|D)$ can be estimated by the stored past trained (fixed) model, because the running average statistics are actually stored in the widely-used BatchNorm layers of the model.

In our proposed method, the technique of Yin et al. (2020) cannot be directly used, because we work in the embedding space. To make it work in our proposed method, we change the structure of the model Yin et al. (2020) and we use a different overall loss function. Our approach involves generating class-conditional images in the embedding space by utilizing the embedding centroid of each class. We adopt minimizing the Euclidean distance between the input noise and the class embedding centroids as one term of the loss function (instead of using the softmax layer and cross-entropy as in Yin et al. (2020)), and the regularization

term in (4) as the second term. Overall, using the fixed model (i.e., the past trained model $M_{k-1}$), the synthetic images $G_k$ are constructed for a target class by the following optimization:

$$\min_{\hat{x}} L_G = \min_{\hat{x}} \left( dist(\hat{x}, \mu) + \lambda_b R_b(\hat{x}) \right), \tag{5}$$

where $\mu$ is the embedding centroid for the target class of the last task, $\lambda_b$ is the scale for the regularization loss, and $dist(i, j)$ is Euclidean distance.

As shown in the past data generator block of Fig. 1, for training the current model $M_k$, the generator is composed of the input image $\hat{x}$ (initialized by the noise) and the fixed (stored) model $M_{k-1}$ that was trained in the last task time $k-1$. In principle, using the model $M_{k-1}$, one might attempt to generate the synthetic images for all past classes from the very first task time to the last task time $k-1$, because ideally $M_{k-1}$ had learned all classes up to task $k-1$. However, it turns out that such an approach does not work well due to *unavoidable* forgetting of the past knowledge.

From our experiments, we found that the best IL performance was achieved when $M_{k-1}$ was used to generate the synthetic images of the last task time $k-1$ only, compared to the case of synthetic images of the last task time $k-1$ and any older tasks. The benefit of generating the synthetic images of the last task time $k-1$ only is that the computational complexity required for generating synthetic past images is substantially lower, which makes our proposed scheme scalable.

Based on our extensive experiments, we found that the subjective image quality of synthetic images is important only up to a certain extent (i.e., truly high-quality images are not required) when using them in our proposed KD method, allowing us to reduce the number of iterations for generating images without significant performance loss. The average accuracy of our proposed IL method versus the number of iterations used for generating past synthetic images for CIFAR-10 is shown in Fig. 5(a). As can be seen, increasing the number of iterations beyond 40 does not further improve the average accuracy of our scheme. In this paper, therefore, we set the number of iterations to 40 for CIFAR-10 and 80 for all the rest datasets (as opposed to 3,000 as in Yin et al. (2020)), of which computing complexity is not high. In Fig. 5(b), forty eight samples of generated synthetic images with a ResNet-18 network trained on CIFAR-10 are illustrated. We consider three different iteration number for generating each 16 synthetic images including 20, 40, and 1000 iterations. In this simulation, our setting is $\lambda_b = 10$ with Adam optimization and the learning rate of 0.06. The superior visual quality of synthetically generated images is apparent with 1000 iterations (bottom 16 images in Fig. 5(b)), surpassing the quality observed with 40 iterations (middle 16 images in Fig. 5(b)). However, it is crucial to note that achieving such high pixel-level quality is not a necessity for our proposed method. Our primary focus lies in capturing the overall distribution within our knowledge distillation approach.

Additionally, our experiments show that the number of required synthetic images is much fewer than the actual images. Based on the experiments, we choose the amount of synthetic data for the last task to be only 25% of the actual data for the current task. It effectively addresses the issue of generation time, which also contributes to training time. To the best of our knowledge, in the literature, no generative replay-based IL method has used such a scalable generating process by re-using a previously stored model both as a lightweight generator and a teacher to distill past knowledge into the current model simultaneously. The impact of varying ratios of synthetic images is presented in Appendix A.1.

### 3.4.2 MSE Regularization Term, $L_{KD}^{mse}$

Using the synthesized past images $G_k$, we introduce a regularization term, which is inspired by self-supervised learning. Minimizing the distance between two embeddings of two differently augmented versions of a single input has proved to be a powerful strategy in self-supervised learning Bardes et al. (2022). Inspired by this, we propose to use a similar loss term, namely, the Mean Square Euclidean (MSE) distance regularization term, $L_{KD}^{mse}$. Our idea is to prevent the knowledge learned in the last task from being forgotten by minimizing the distance between two embeddings produced respectively by the last model $M_{k-1}$ (teacher) and current model $M_k$ (student) for the same last data $G_k$. Specifically, we supply the same input $G_k$ (i.e., the synthetic images for the last task) to the current trainable model $M_k$ and the stored (fixed) last model $M_{k-1}$. For training of $M_k$, we then minimize the distance between the two sets of embeddings produced by those two

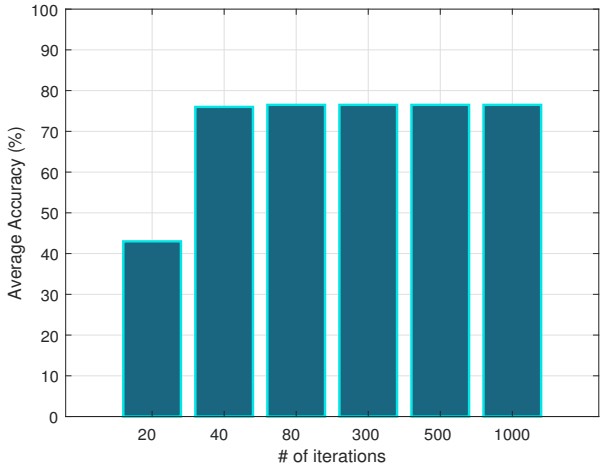
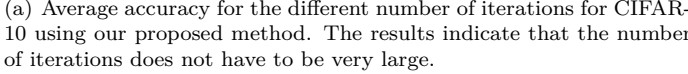
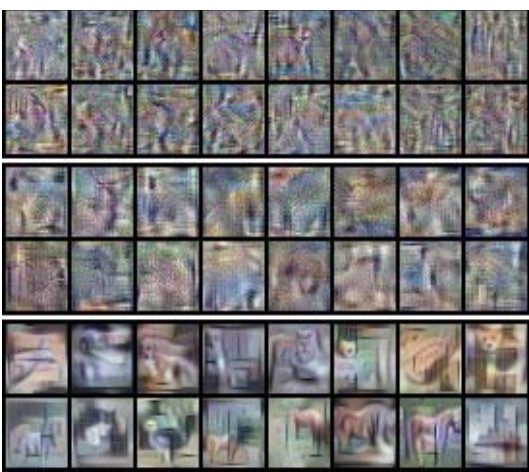

(a) Average accuracy for the different number of iterations for CIFAR-10 using our proposed method. The results indicate that the number of iterations does not have to be very large.

(b) Forty eight generated synthetic images (32×32) for three different number of iterations: (i) 20 iterations is top 16 images, (ii) 40 iterations is middle 16 images, and (iii) 1000 iterations is bottom 16 images. A ResNet-18 is trained on CIFAR-10.

Figure 5: Synthetic images versus the number of iterations.

models. This way, in the embedding space, the current model $M_k$ will learn the knowledge of the embedding positions that had been learned by $M_{k-1}$ for the images $G_k$ of the last task $k-1$.

To mathematically formulate the loss, let $\tilde{Z}_k = [\tilde{z}_k^1, \ldots, \tilde{z}_k^n] \in \mathbb{R}^{p \times n}$ denote the un-normalized embedding matrix produced in the current task time $k$ by the trainable model $M_k$ when the inputs are the synthetic images, $G_k$. Similarly, $\tilde{Z}'_k = [\tilde{z}'^1_k, \ldots, \tilde{z}'^n_k] \in \mathbb{R}^{p \times n}$ represents the un-normalized embedding matrix produced in the current task time $k$ by the fixed last model $M_{k-1}$ when the inputs are the same synthetic images, $G_k$. The vector $\tilde{z}_k^j$ is the $p$-dimensional un-normalized embedding vector obtained by model $M_k$ when the input is the $j^{\text{th}}$ synthetic image $x_j \in G_k$ in a mini-batch composed of $n$ images. The column vector $\tilde{z}'^j_k$ is defined in the same way, but by $M_{k-1}$. Our proposed regularization term is the element-wise MSE distance between the two embedding matrices, $L_{KD}^{mse} = \frac{1}{n}\|\tilde{Z}_k - \tilde{Z}'_k\|^2 = \frac{1}{n}\sum_{j=1}^n \| \tilde{z}_k^j - \tilde{z}'^j_k \|^2$.

Alternatively, one might try to use different distance measures such as mean absolute error (MAE) or cosine dissimilarity losses. However, based on our simulations, MSE exhibited the best performance. For the detailed simulation results, see Fig. 9 in Section 4.4.

### 3.4.3 Intra-class and Inter-class Regularization Terms, $L_{KD}^{intra}$ and $L_{KD}^{inter}$

In order to further mitigate the catastrophic forgetting, we propose two additional loss terms, through which the past model $M_{k-1}$ teaches $M_k$ how to *not* forget the knowledge of the past data in the embedding space, as illustrated in Fig. 6. Specifically, the intra-class loss and the inter-class loss are designed respectively by constructing intra-class and inter-class embedding scatters at the level of each mini-batch. Let $\mathcal{G}_{i,k} \subseteq G_k$ denote the subset of $G_k$ that contains the synthetic images of the $i^{\text{th}}$ class only (in the last task), $i = 1, \ldots, C$, where $C$ is the number of classes in $G_k$ (i.e., the number of classes for the last task, $k-1$). Letting $|\mathcal{G}_{i,k}|$ denote the number of images in $\mathcal{G}_{i,k}$, we use $Z_{i,k} = [z_{i,k}^1, \ldots, z_{i,k}^{|\mathcal{G}_{i,k}|}] \in \mathbb{R}^{p \times |\mathcal{G}_{i,k}|}$ to represent the normalized embedding matrix produced by the current model $M_k$ when the inputs are the synthetic images in $\mathcal{G}_{i,k}$. The vector $z_{i,k}^j$ represents the $p$-dimensional normalized embedding vector produced by $M_k$ when the input is the $j^{\text{th}}$ image of class $i$ in $\mathcal{G}_{i,k}$. The class mean of the embedding vectors $z_{i,k}^j$ for class $i$ is determined by: $\mu_{i,k} = \frac{1}{|\mathcal{G}_{i,k}|}\sum_{j=1}^{|\mathcal{G}_{i,k}|} z_{i,k}^j$. We will construct scatter vectors at the level of mini-batches, each of which is of size

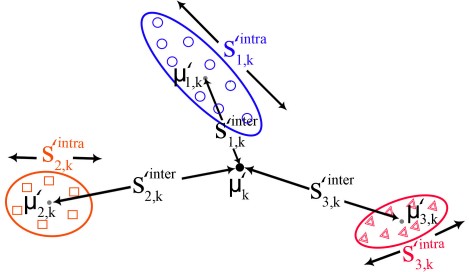
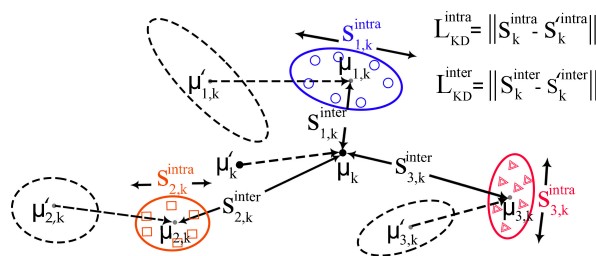

(a) Embeddings produced (in the current task time $k$) by $M_{k-1}$ using the synthetic images $G_k$ for the three classes of the last task, $k-1$. Intra-class scatter matrices $S_{i,k}^{'intra}$ and inter-class scatter matrices $S_{i,k}^{'inter}$ are shown.

(b) Embeddings produced (in the current task time $k$) by $M_k$ using the synthetic images $G_k$ for the three classes of the last task, $k-1$. Intra-class scatter matrices $S_{i,k}^{intra}$ and inter-class scatter matrices $S_{i,k}^{inter}$ are shown. To prevent forgetting the knowledge learned by $M_{k-1}$, we minimize the displacements from $S_k^{'intra}$ to $S_k^{intra}$, and from $S_k^{'inter}$ to $S_k^{inter}$, over two consecutive task times.

Figure 6: Knowledge distillation through intra-class and inter-class losses.

$n$. Recall that each mini-batch contains the synthetic images only for $c$ classes (not $C$). Then the mini-batch mean over $c$ classes is given by $\mu_k = \frac{1}{c} \sum_{i=1}^{c} \mu_{i,k}$.

At the mini-batch level, we now define the inter-class scatter $S_{i,k}^{inter}$ in the current task time $k$, which is constructed from the outputs of $M_k$, for the synthetic images $G_k$ of the classes in the last task $k-1$ as $S_{i,k}^{inter} = (\mu_{i,k} - \mu_k), i = 1, \ldots, c$. In the same manner, $S_{i,k}^{'inter}$ is defined as the inter-class scatter, which is constructed in the current task time $k$ from the outputs of $M_{k-1}$, for the images $G_k$ of the classes in the last task $k-1$. In order to transfer the knowledge from $M_{k-1}$ (teacher) to $M_k$ (student) about the embedding cluster distances among different classes learned in the last task $k-1$, we design the inter-class regularization term $L_{KD}^{inter}$ by penalizing the change from $S_{i,k}^{'inter}$ to $S_{i,k}^{inter}$ as follows:

$$L_{KD}^{inter} = \sum_{i=1}^{c} \parallel S_{i,k}^{inter} - S_{i,k}^{'inter} \parallel. \tag{6}$$

The second part of the information that we try to keep unchanged in learning the $k^{\text{th}}$ task is the intra-class scatter denoted as $S_{i,k}^{intra}$. The $S_{i,k}^{intra}$ is constructed in the current task time $k$ from the outputs of $M_k$, for the images $G_k$ of the classes in the last task $k-1$: $S_{i,k}^{intra} = \sum_{j=1}^{|\mathcal{G}_{i,k}|} (z_{i,k}^j - \mu_{i,k}), i = 1, \ldots, c$.

Similarly, $S_{i,k}^{'intra}$ is defined as the intra-class scatter, constructed in the current task time $k$ from the outputs of $M_{k-1}$, for the images $G_k$ of the classes in the last task $k-1$. Our goal in the intra-class loss, denoted as $L_{KD}^{intra}$, is to penalize the change from $S_{i,k}^{'intra}$ to $S_{i,k}^{intra}$ in order to transfer the knowledge (from $M_{k-1}$ to $M_k$) about the embeddings within each class learned in the last task $k-1$.

$$L_{KD}^{intra} = \sum_{i=1}^{c} \parallel S_{i,k}^{intra} - S_{i,k}^{'intra} \parallel. \tag{7}$$

As shown in (1), the final training mini-batch-wise loss $L_{M_k}$ is the combination of the triplet loss, $L_{tri}$, inter-task confusion regularizer, $L_{TCR}$, and the KD loss composed of the three proposed KD loss terms $L_{KD}^{mse}$, $L_{KD}^{intra}$, and $L_{KD}^{inter}$ as in (3).

## 4 Experiments

### 4.1 Experimental Setting

**Datasets**: The five most widely used datasets are selected. **MNIST** Deng (2012) and **CIFAR-10** Krizhevsky (2009) includes ten classes, which are divided into five disjoint tasks, resulting in two classes

per task. **Mini-ImageNet** Vinyals et al. (2016) is composed of 100 classes, and following Gu et al. (2022), we split 100 classes into ten disjoint tasks, including ten classes in each task. **CIFAR-100** Krizhevsky (2009) has 100 classes and two methods for splitting this dataset are used: (i) the entire dataset is divided into ten disjoint tasks, with 10 classes per task, (ii) half of the classes are used for the first task, and the remaining half classes for the rest phases as in Zhu et al. (2021; 2022). **TinyImageNet** Le & Yang (2015) contains 200 classes and half of the classes are used for the first task as in Zhu et al. (2021; 2022).

**Baselines:** For performance comparison, we first consider a simple method of fine-tuning only the current model using the conventional triplet loss, which can be considered as a performance lower bound. We also compare our method with the following state-of-the-arts (SOTAs): **LwF** Li & Hoiem (2018), **EWC++** Chaudhry et al. (2018a), **AGEM** Chaudhry et al. (2018b), **ER** Chaudhry et al. (2019), **GSS** Aljundi et al. (2019b), **MIR** Aljundi et al. (2019a), **ASER_$\mu$** Shim et al. (2021), **SCR** Mai et al. (2021), **Gen.** van de Ven et al. (2021), **CF-IL** PourKeshavarzi et al. (2022), **Semi** Michel et al. (2022), **MGI** Gu et al. (2022), **SDC** Yu et al. (2020), **Pass** Zhu et al. (2021), and **S-SRE** Zhu et al. (2022), **ES-MER** Sarfraz et al. (2023), **IMEX-Reg** Bhat et al. (2024), **CW-DPPER** Chen et al. (2024), **LODE** Liang & Li (2024).

**Implementation details**: As the trainable encoder, $M_k$, we use a ResNet-18 with the classifier part and the fully-connected layer removed as in Mai et al. (2021). Therefore, the embedding dimension $p$ is equal to 512. All the baseline methods except the Gen. van de Ven et al. (2021) and CF-IL PourKeshavarzi et al. (2022) are evaluated with NCM. To comply with the majority of the baselines, Adam is used as the optimizer with a learning rate 1e−6 and weight decay of 0.0001. The total number of epochs is 50 for each task. The mini-batch size $b$ for constructing the triplet loss $L_{tri}$ and the inter-task confusion regularizer $L_{TCR}$ is $b = 64$. The mini-batch size $n$ for constructing the three KD losses is $n = 16$. This means that, for constructing the total loss $L_{M_k}$ in each mini-batch, the total number of synthetic images is one-fourth of the current (new) actual images. Every individual mini-batch (whether from $D_k$ or $G_k$) is set to contain the samples only for $c$ different classes, where $c = 2$ for CIFAR-10 and MNIST. For CIFAR-100, Mini-ImageNet, and TinyImageNet we set $c = 4$. This means that each mini-batch of size $b = 64$ drawn from $D_k$ contains 32 actual images for each class of both CIFAR-10 and MNIST, 16 actual images for each class of CIFAR-100, Mini-ImageNet, and TinyImageNet. Also, each mini-batch of size $n = 16$ drawn from $G_k$ contains 8 synthetic images for each class of CIFAR-10 and MNIST, 4 synthetic images for each class of CIFAR-100, Mini-ImageNet, and TinyImageNet. Hyper-parameters in our method are: $\lambda_1 = 0.2$, $\lambda_2 = 0.8$, $\lambda_3 = 0.4$, and $\lambda_b = 10$ for CIFAR-10, MNIST, and CIFAR-100. For Mini-ImageNet and TinyImageNet we set $\lambda_1 = 0.1$, $\lambda_2 = 0.3$, $\lambda_3 = 0.2$, and $\lambda_b = 15$ (see Appendix A.1 for the details).

Learning of the initial task in IL is non-incremental since there is nothing before it. Therefore, we use softmax loss only in the initial non-incremental step because softmax proved its power in static offline learning. For the non-incremental step, we use SGD with a learning rate of 0.001, momentum of 0.9, weight decay of 0.0005, and mini-batch size of 256 for all datasets. After completing the training of the initial non-incremental step, we remove the classifier part and use only the encoder part in the remaining incremental tasks. As a regularization method, we add some controlled noise to the triplets in the embedding space, akin to classical regularization methods, with the goal of reducing overfitting. We refer to this technique as "noisy triplet loss" in our simulation results (see Appendix A.3 for the details).

## 4.2 Comparison Results

In terms of average accuracy $A_K$, our proposed method is compared to all mentioned baselines in the class-IL setting. Table 1 presents the experimental findings pertaining to equal splitting datasets, which means the number of classes is the same in all tasks. We used the repository [1] to replicate the results for the baseline methods: LwF Li & Hoiem (2018), EWC++ Chaudhry et al. (2018a), AGEM Chaudhry et al. (2018b), ER Chaudhry et al. (2019), GSS Aljundi et al. (2019b), MIR Aljundi et al. (2019a), ASER_$\mu$ Shim et al. (2021), and SCR Mai et al. (2021), within a controlled experimental framework. In all experiments, the data splits and the random seeds used for class selection were consistent across trials, with the reported results representing the average of 10 different seeds (task orderings). Additionally, we reproduced the average

---

[1]`https://github.com/RaptorMai/online-continual-learning`

accuracy shown in Table 1 for the methods ESMER Sarfraz et al. (2023), IMEX-Reg Bhat et al. (2024), CW-DPPER Chen et al. (2024), and LODE Liang & Li (2024), using the same data splits and task orderings. However, for the methods Gen. van de Ven et al. (2021),CF-IL PourKeshavarzi et al. (2022), Semi Michel et al. (2022), andMGI Gu et al. (2022), we were unable to replicate the results based on the available source code; consequently, we extracted the results directly from the respective original papers. Symbol † indicates that the numerical results are directly copied from the original papers and '⋆' denotes that the results are not available in those original papers. Whether each method stores any *actual* past data samples or not is denoted by ‡ and ⋄, where ⋄ indicates that no actual past data samples are stored and ‡ represents the coreset replay-based methods. These results have been compared with the baselines that adopt the same data-splitting approach.

The results of our proposed method were derived from ten independent runs with ten different class orderings. From the results, we can see that our proposed method outperforms all baselines by notable margins, achieving the state-of-the-art (SOTA) performance. Even compared to Gen. van de Ven et al. (2021) which requires multiple models as many as the number of classes (e.g., 100 models for CIFAR-100), our method works better despite that we use only two models, (trainable) $M_k$ and (stored and fixed) $M_{k-1}$, irrespective of the total number of classes. The memory requirements for each method are also provided to offer better insights.

Table 2 presents the findings regarding an increased number of tasks and the comparison with prior research that utilized half of the classes in the initial task. Notably, the results show that our proposed method outperforms previous approaches by a significant margin as the number of tasks increases. Although most Class-IL studies use average accuracy as their primary metric, we also analyze final task accuracy, denoted as $A_{last}$, in Table 3 for the same number of classes in all tasks setting to provide additional insights. This comparison reveals that our method generally outperforms existing approaches in final task performance.

Table 1: Average accuracy, $A_K$ (%), after finishing the final task. Our method considerably outperforms all methods, including the coreset replay-based methods on all three datasets. The 'MS-10', 'MS-100', and 'MS-Mini' denote the memory size (MB) for CIFAR-10, CIFAR-100, Mini-ImageNet respectively.

| Schemes | MNIST | CIFAR-10 | CIFAR-100 | Mini-ImageNet | MS-10 | MS-100 | MS-Mini |
|---------|-------|----------|-----------|---------------|-------|--------|---------|
|         | $K = 5$ | $K = 5$ | $K = 10$ | $K = 10$ |  |  |  |
| Joint training⋄ | $98.1_{\pm0.06}$ | $83.9_{\pm0.6}$ | $74.3_{\pm1.2}$ | $72.1_{\pm0.2}$ | ⋆ | ⋆ | ⋆ |
| Fine-tune⋄ | $18.2_{\pm0.1}$ | $16.9_{\pm1.1}$ | $5.4_{\pm0.9}$ | $4.3_{\pm0.9}$ | 42.6 | 42.6 | 42.6 |
| LwF⋄ | $23.2_{\pm0.1}$ | $19.3_{\pm0.4}$ | $13.8_{\pm0.4}$ | $8.6_{\pm0.5}$ | 42.6 | 42.6 | 42.6 |
| EWC++ ⋄ | $55.2_{\pm0.1}$ | $17.9_{\pm0.3}$ | $5.7_{\pm0.2}$ | $4.2_{\pm0.3}$ | 140.4 | 140.6 | 140.6 |
| AGEM‡ | $64.7_{\pm0.3}$ | $27.8_{\pm1.2}$ | $14.3_{\pm0.3}$ | $11.7_{\pm0.5}$ | 108.9 | 108.9 | 199.4 |
| ER‡ | $67.1_{\pm1.1}$ | $47.1_{\pm0.9}$ | $27.6_{\pm1.1}$ | $20.8_{\pm1.2}$ | 46.9 | 48.4 | 57.4 |
| GSS‡ | $69.8_{\pm0.2}$ | $46.7_{\pm1.6}$ | $26.8_{\pm0.4}$ | $21.0_{\pm1.1}$ | 95.1 | 95.1 | 104.2 |
| MIR‡ | $71.8_{\pm0.30}$ | $49.3_{\pm1.2}$ | $27.3_{\pm1.0}$ | $21.8_{\pm1.0}$ | 95.1 | 108.9 | 198.4 |
| $\text{ASER}_\mu$ ‡ | $75.5_{\pm0.2}$ | $50.3_{\pm1.0}$ | $29.3_{\pm0.5}$ | $22.1_{\pm0.2}$ | 62.1 | 62.1 | 152.6 |
| SCR‡ | $79.2_{\pm0.2}$ | $65.7_{\pm0.6}$ | $37.5_{\pm0.5}$ | $35.3_{\pm0.3}$ | 48.3 | 52.9 | 89.1 |
| Gen.⋄ | $93.79_{\pm0.08}$ † | $56.0_{\pm0.04}$ † | $49.5_{\pm0.06}$ † | ⋆ | 99.2 | 99.2 | ⋆ |
| CF-IL⋄ | $95.3_{\pm0.15}$ | 75.34 † | ⋆ | ⋆ | 59.2 | ⋆ | ⋆ |
| Semi‡ | $92.3_{\pm0.10}$ | $57.9_{\pm1.1}$ † | $38.9_{\pm0.5}$ † | ⋆ | 96.8 | 102.3 | ⋆ |
| MGI‡ | $90.8_{\pm0.30}$ | $52.1_{\pm2.5}$ † | $24.1_{\pm0.8}$ † | $19.1_{\pm0.9}$ † | 96.2 | 99.8 | 135.4 |
| ESMER‡ | $93.5_{\pm0.2}$ | $71.2_{\pm0.4}$ | $50.6_{\pm0.4}$ | $45.9_{\pm1.1}$ | 94.2 | 94.2 | 97.8 |
| IMEX-Reg‡ | $95.7_{\pm0.10}$ | $74.7_{\pm0.1}$ | $49.9_{\pm0.3}$ | $46.1_{\pm0.9}$ | 98.9 | 98.9 | 102.5 |
| CW-DPPER‡ | $91.8_{\pm0.2}$ | $67.8_{\pm0.8}$ | $49.6_{\pm0.5}$ | $49.8_{\pm0.1}$ | 99.7 | 99.7 | 135.9 |
| LODE‡ | $95.5_{\pm0.9}$ | $76.5_{\pm0.7}$ | $51.6_{\pm0.9}$ | $52.9_{\pm0.4}$ | 49.7 | 63.9 | 59.7 |
| **Our method⋄** | $\mathbf{96.1}_{\pm0.8}$ | $\mathbf{76.9}_{\pm0.8}$ | $\mathbf{54.8}_{\pm0.9}$ | $\mathbf{54.2}_{\pm1.2}$ | 50.4 | 53.4 | 59.2 |

Evaluating forgetting $O_k$ serves as an additional measure to assess the extent to which a model loses information after learning new knowledge. The lower forgetting values are the lesser forgetting concerning earlier tasks. The results of average accuracy and forgetting after each task are summarized in Table 4. We can see a huge margin between fine-tuning and our method regarding forgetting values in all three datasets.

To gain deeper insights into the model's performance on individual tasks, we generated confusion matrices for each task for CIFAR-10 with 5 equal tasks. These matrices provide a visual representation of the model's

Table 2: Average accuracy, $A_K$ (%), after finishing the final task setting half of the classes for the first task.

| Schemes | CIFAR-100 | | | TinyImageNet | | |
|---|---|---|---|---|---|---|
| | $K = 5$ | $K = 10$ | $K = 20$ | $K = 5$ | $K = 10$ | $K = 20$ |
| Joint training⋄ | 74.32 | 74.32 | 74.32 | 66.81 | 66.81 | 66.81 |
| SDC Yu et al. (2020)⋄ | 56.77† | 57.00† | 58.90† | ⋆ | ⋆ | ⋆ |
| Pass Zhu et al. (2021)⋄ | 63.47† | 61.84† | 58.09† | 49.55† | 47.29† | 42.07† |
| S-SRE Zhu et al. (2022)⋄ | 65.88† | 65.04† | 61.70† | 50.39† | 48.93.1† | 48.17† |
| LODE Liang & Li (2024)⋄ | 65.88† | 65.04† | 61.70† | 50.39† | 48.93† | 48.17† |
| **Our method⋄** | $\mathbf{66.15}_{\pm 0.10}$ | $\mathbf{65.82}_{\pm 0.30}$ | $\mathbf{64.35}_{\pm 0.70}$ | $\mathbf{55.32}_{\pm 1.2}$ | $\mathbf{54.90}_{\pm 1.5}$ | $\mathbf{54.10}_{\pm 1.4}$ |

Table 3: Final task accuracy, $A_{last}$ (%).

| Schemes | CIFAR-100 | | | TinyImageNet | | |
|---|---|---|---|---|---|---|
| | $K = 5$ | $K = 10$ | $K = 20$ | $K = 5$ | $K = 10$ | $K = 20$ |
| Joint training⋄ | 74.32 | 74.32 | 74.32 | 66.81 | 66.81 | 66.81 |
| SCR Mai et al. (2021)‡ | 31.22 | 28.65 | 26.10 | 29.15 | 27.45 | 23.11 |
| ESMER Sarfraz et al. (2023)‡ | 49.90† | 48.77 | 47.35 | 48.85 | **47.30** | 48.40 |
| IMEX-Reg Bhat et al. (2024)‡ | 50.10 | 48.54 | 46.25 | 49.65 | 46.64 | 41.34 |
| LODE Liang & Li (2024)⋄ | 51.00 | 46.31 | 46.00 | 50.20 | 46.80 | **45.11** |
| **Our method⋄** | **51.25** | **49.67** | **47.83** | **50.76** | 47.13 | 44.85 |

Table 4: Forgetting (%) and average accuracy (%) after each task for our method, LODE, and the lower bound (i.e., fine-tuning).

| Method | Metric | CIFAR-10 | | | | | | | | |
|---|---|---|---|---|---|---|---|---|---|---|
| | $k$ | 1 | 2 | 3 | 4 | - | - | - | - | - |
| Fine tuning | Forgetting | 9.5 | 12.2 | 15.7 | 28.6 | - | - | - | - | - |
| | Accuracy | 50.9 | 40.0 | 30.0 | 16.9 | - | - | - | - | - |
| LODE Liang & Li (2024) | Forgetting | 4.2 | 8.4 | 5.1 | 9.4 | - | - | - | - | - |
| | Accuracy | 90.5 | 81.9 | 79.8 | 75.4 | - | - | - | - | - |
| Our method | Forgetting | 4.6 | 4.7 | 5.4 | 11.1 | - | - | - | - | - |
| | Accuracy | 88.1 | 83.3 | 81.1 | 75.9 | - | - | - | - | - |
| **Method** | **Metric** | **CIFAR-100** | | | | | | | | |
| | $k$ | 1 | 2 | 3 | 4 | 5 | 6 | 7 | 8 | 9 |
| Fine tuning | Forgetting | 3.8 | 12.1 | 7.0 | 7.9 | 10.2 | 11.4 | 5.9 | 4.3 | 4.0 |
| | Accuracy | 69.8 | 54.4 | 49.1 | 41.8 | 32.2 | 20.0 | 14.6 | 9.1 | 6.5 |
| LODE Liang & Li (2024) | Forgetting | 5.0 | 10.0 | 9.2 | 2.7 | 1.5 | 1.2 | 1.2 | 1.7 | 9.4 |
| | Accuracy | 86.3 | 74.1 | 67.3 | 64.7 | 62.8 | 61.1 | 60.4 | 58.4 | 51.5 |
| Our method | Forgetting | 5.3 | 8.6 | 3.2 | 2.9 | 2.6 | 2.2 | 1.0 | 1.7 | 4.1 |
| | Accuracy | 82.2 | 71.8 | 69.5 | 66.7 | 63.4 | 60.2 | 59.6 | 57.5 | 54.5 |
| **Method** | **Metric** | **Mini-ImageNet** | | | | | | | | |
| | $k$ | 1 | 2 | 3 | 4 | 5 | 6 | 7 | 8 | 9 |
| Fine tuning | Forgetting | 12.1 | 18.9 | 14.5 | 5.3 | 9.3 | 4.8 | 5.4 | 3.5 | 2.1 |
| | Accuracy | 51.5 | 32.1 | 20.0 | 18.9 | 11.0 | 9.8 | 6.1 | 5.5 | 5.2 |
| LODE Liang & Li (2024) | Forgetting | 8.7 | 9.7 | 4.8 | 3.4 | 3.0 | 3.1 | 2.0 | 9.7 | 9.1 |
| | Accuracy | 79.3 | 70.8 | 64.4 | 62.6 | 59.2 | 56.0 | 54.1 | 50.0 | 48.80 |
| Our method | Forgetting | 11.1 | 6.2 | 3.4 | 3.7 | 2.2 | 2.3 | 2.0 | 4.2 | 6.6 |
| | Accuracy | 76.8 | 71.4 | 66.8 | 64.8 | 62.3 | 59.9 | 58.0 | 56.2 | 55.3 |

classification accuracy across different classes within each task. Fig. 7 shows the confusion matrices for all 5 tasks, where $U_{j,k}$ presents the confusion matrix obtained from model $M_k$, which was trained on task $k$ and then tested on classes for tasks 0 to $j$, $j \leq k$. By examining the confusion matrices, we can see the patterns of misclassification arising from both catastrophic forgetting and task-confusion. Table 5 illustrates the order of

classes for each task. From the visualization in Fig. 7, it is evident that the highest correct classification for $U_{j,k}$ occurs when $k = j$. However, it is noteworthy that the worst classification is not necessarily associated with the initial classes (i.e., $j \ll k$) suggesting the significance of the class ordering.

Table 5: The labels of classes associated with each task. Classes marked in bold represent the classes present at that task number. For instance, in task $k = 2$, the current classes are $\{0, 3\}$, and previous classes are $\{4, 2, 7, 6\}$.

| Task# (k) | Categories | | | | | | | | | |
|---|---|---|---|---|---|---|---|---|---|---|
| 0 | **4** | **2** | - | - | - | - | - | - | - | - |
| 1 | 4 | 2 | **7** | **6** | - | - | - | - | - | - |
| 2 | 4 | 2 | 7 | 6 | **0** | **3** | - | - | - | - |
| 3 | 4 | 2 | 7 | 6 | 0 | 3 | **5** | **8** | - | - |
| 4 | 4 | 2 | 7 | 6 | 0 | 3 | 5 | 8 | **9** | **1** |

It can be clearly seen that controlling the position of embeddings in the embedding space through our proposed losses is very effective for mitigating inter-task confusion and combating catastrophic forgetting, which leads to superior performance to all those baselines. Furthermore, our results echo the claim of Mai et al. (2021); Yu et al. (2020) that, in IL, using metric learning with NCM classifiers could be a better choice than softmax classifiers.

## 4.3 Ablation Study on $L_{M_k}$

In this subsection, more experiments are conducted to reveal the effect of each term of our loss $L_{M_k}$ in (1). We present the results of ablation experiments for inter-task confusion regularization, knowledge distillation (MSE, intra-class, and inter-class), and their collective impact on four distinct datasets. With the exception of TinyImageNet, where we allocate half of the total classes to the initial non-incremental task, all other datasets are configured with an equal distribution of classes in each task.

**Effect of inter-task confusion regularization, $L_{TCR}$:** To gauge the impact of $L_{TCR}$ on the performance, we set $\lambda_1 = 0$. The intention behind incorporating this term $L_{TCR}$ is to prevent overlap across the current classes and those classes that have been learned earlier, thereby mitigating the occurrence of inter-task confusion. The average accuracy is represented in the third row of Table 6. Looking at Table 6, it becomes evident that eliminating the inter-task confusion regularizer leads to a decline of no less than 1.1% across all datasets. Further insight into the impact of this term on individual intermediate tasks is provided in Fig. 8 (depicted by the yellow curve), illustrating a heightened prominence of the influence of $L_{TCR}$ as the number of tasks increases.

**Effect of the MSE loss, $L_{KD}^{mse}$:** To assess how $L_{KD}^{mse}$ affects the performance, we assign a value of $\lambda_2 = 0$, (i.e., total loss is $L_{M_k} = L_{tri} + \lambda_1 L_{TCR} + \lambda_3 (L_{KD}^{intra} + L_{KD}^{inter})$). The objective of this term is to minimize the Euclidean distance between the embeddings of the current under-training model, denoted as $M_k$ (the student), and the previously stored fixed model $M_{k-1}$ (the teacher) when the input comprises data generated from a prior task. This approach compels $M_k$ to acquire knowledge about past tasks from the teacher model $M_{k-1}$, thereby mitigating the phenomenon of catastrophic forgetting.

The fourth row of Table 6 displays the performance after completing the final task across all datasets. Upon comparison with the outcomes of the $L_{TCR}$, it becomes evident that the influence of this term is more pronounced. Removing it results in a performance decrease of around 3%. Additionally, in Fig. 8, a comparison is made between employing the MSE loss term and not using it (i.e., setting $\lambda_2 = 0$) for each incremental task. The significance of the MSE loss becomes evident in the outcomes as the number of tasks increases.

**Effect of the intra- and inter-class losses, $L_{KD}^{intra}$ and $L_{KD}^{inter}$:** We set $\lambda_3 = 0$, to see the impact of our proposed intra- and inter-class losses on the average accuracy. The fifth row of Table 6 displays the results of the performance evaluation following the completion of the final task. It is evident that intra- and inter-class losses play a pivotal role in our suggested loss function. Its significance is underscored by a reduction of up to 10% in the average accuracy upon its removal. This term embodies our key strategy for

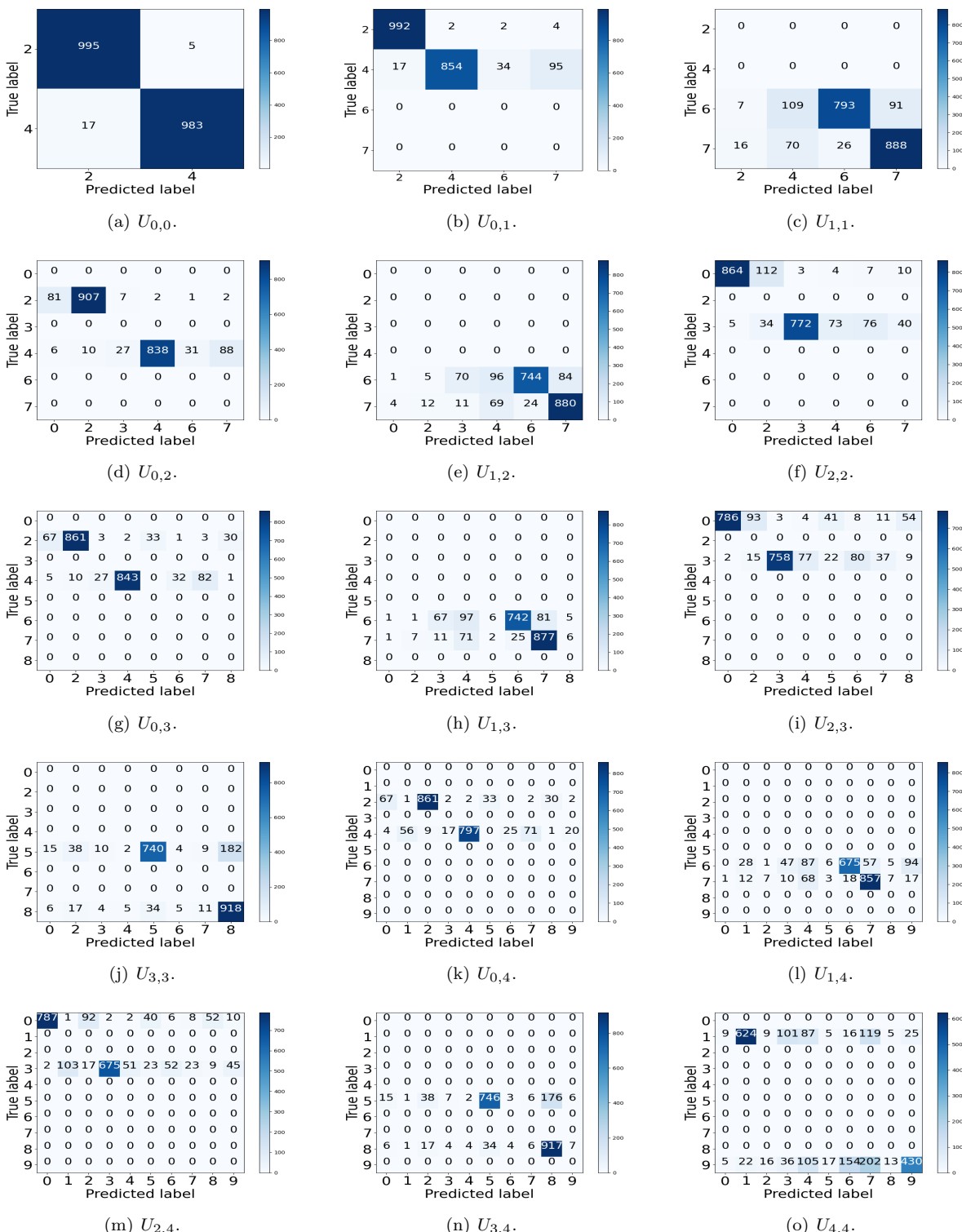

Figure 7: Confusion matrix $U_{j,k}, j = 0, \ldots, 4, k = 0, \ldots, 4$, using CIFAR-10

mitigating catastrophic forgetting, achieved through the innovative preservation of the positional relationship of previously learned classes within the embedding space. Moreover, looking at Fig. 8, the drop in the average

Table 6: Average accuracy, $A_K$ (%), after finishing the final task for ablation experiments. The mean and standard deviation (STD) are reported from ten individual runs with ten different orderings of the classes.

| Schemes | CIFAR-10 | CIFAR-100 | Mini-ImageNet | TinyImageNet |
|---|---|---|---|---|
| | $K = 5$ | $K = 10$ | $K = 10$ | $K = 20$ |
| Fine-tune | $24.7_{\pm 0.8}$ | $37.2_{\pm 1.2}$ | $10.0_{\pm 0.85}$ | $6.5_{\pm 1.1}$ |
| Our method when $\lambda_1 = 0$ | $75.7_{\pm 1.2}$ | $52.8_{\pm 0.8}$ | $53.1_{\pm 1.1}$ | $53.0_{\pm 0.95}$ |
| Our method when $\lambda_2 = 0$ | $74.8_{\pm 0.92}$ | $49.5_{\pm 0.85}$ | $51.0_{\pm 1.0}$ | $51.1_{\pm 0.9}$ |
| Our method when $\lambda_3 = 0$ | $70.1_{\pm 0.9}$ | $44.8_{\pm 0.8}$ | $49.1_{\pm 1.2}$ | $48.4_{\pm 0.95}$ |
| Our method | $76.9_{\pm 0.8}$ | $54.8_{\pm 0.9}$ | $54.2_{\pm 1.2}$ | $54.1_{\pm 1.4}$ |

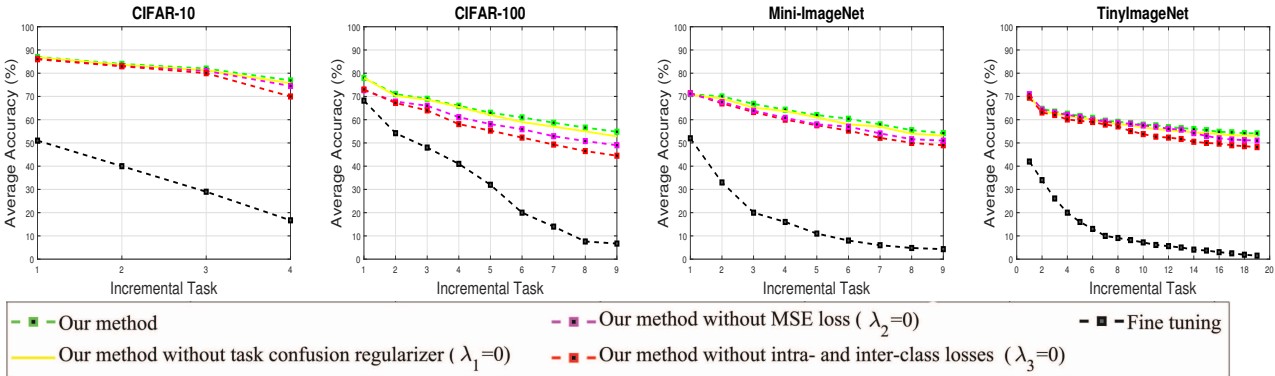

Figure 8: Average accuracy, $A_k$ (%), for ablation experiments for all four datasets (best viewed in color). Only the incremental tasks (or states) are plotted.

accuracy is noticeable for each incremental task when $\lambda_3 = 0$, which means the two losses play an essential role in all datasets.

## 4.4 Ablation Study on MSE Loss, $L_{mse}$

This section delves into the effects of various loss configurations on Mean Square Euclidean (MSE) loss. To assess this, we substituted mean absolute error (MAE) and cosine dissimilarity for MSE loss in our KD loss framework, evaluating their impact on CIFAR-100 and TinyImageNet with a total number of tasks $K = 10$. Fig. 9 illustrate the outcomes, indicating that MSE consistently produces the optimal results. However, it is noteworthy that even with alternative configurations, our method surpasses most state-of-the-art. Thus, while our approach performs admirably across different loss setups, a meticulously crafted configuration can lead to a notable performance enhancement as evidenced in our ablation study.

## 4.5 Ablation Analysis of CF and ITC to Performance Degradation

In this subsection, we aim to analyze the extent of performance reduction caused by CF and ITC independently. CF refers to the phenomenon in continual learning where a model, when trained sequentially on new tasks, experiences a significant decline in performance on previously learned tasks. This occurs as the parameters optimized for new tasks overwrite or interfere with the representations learned for earlier tasks. In contrast, ITC arises when the model struggles to differentiate between tasks, resulting in incorrect predictions that combine knowledge from multiple tasks. While CF primarily affects the retention of prior knowledge, ITC underscores difficulties in task discrimination. Thus, the proportion of ITC can be quantified using the following metric:

$$ITC = (\text{Accuracy given task-IDs}) - (\text{Accuracy without task-IDs}). \tag{8}$$


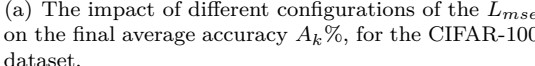
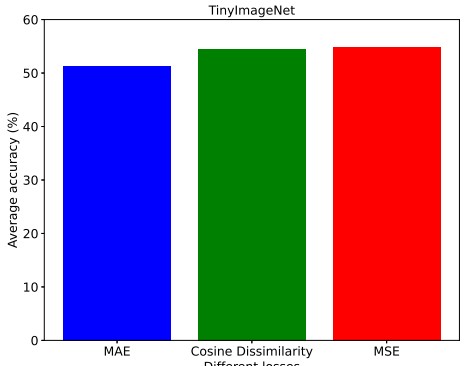

(a) The impact of different configurations of the $L_{mse}$ on the final average accuracy $A_k\%$, for the CIFAR-100 dataset.

(b) The impact of different configurations of the $L_{mse}$ on the final average accuracy $A_k\%$, for the TinyImageNet dataset.

Figure 9: The effects of various loss configurations on MSE loss.

Fig. 10 illustrates the performance degradation caused by CF and ITC for our method, compared to fine-tuning and two recent approaches. As shown, our method effectively mitigates both ITC (indicated by the red values next to the red arrows) and CF (indicated by the black values next to the black arrows) more efficiently than the compared methods.

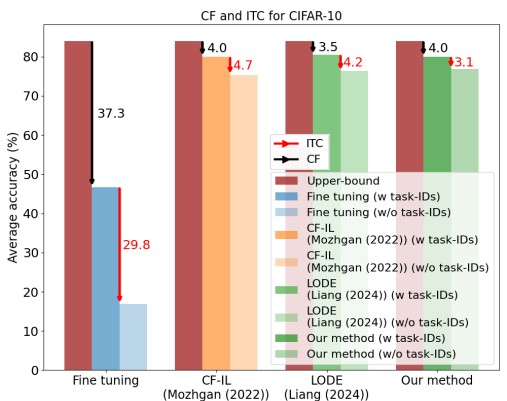
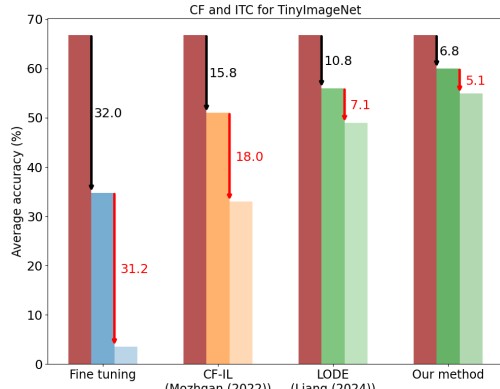

(a) The values of CF and ITC on the final average accuracy $A_k\%$, for the CIFAR-10 dataset.

(b) The values of CF and ITC on the final average accuracy $A_k\%$, for the TinyImageNet dataset.

Figure 10: Analysis of CF and ITC to Performance Degradation (legend provided in (a)).

## 4.6 Ablation Study on Architectures

For further experimental results, we investigated additional network architectures, as shown in Table 7. From the table, it is evident that as the size of the network increases, the average accuracy improves. However, the improvement observed for CIFAR-100 is less pronounced compared to TinyImageNet.

## 4.7 Ablation Study on Memory-Agnostic Measure

In this subsection, we present results for a memory-agnostic measure proposed by Zhou et al. (2024) and Zhou et al. (2023). This metric calculates the area under the performance-memory curve (AUC). To evaluate this metric, we computed the average accuracy of various methods across a range of memory sizes, from small

Table 7: Average accuracy, $A_K$ (%) for different networks.

| Architecture | CIFAR-100 | TinyImageNet |
|---|---|---|
| | $K = 10$ | $K = 20$ |
| ResNet-18 | $65.82_{\pm 0.3}$ | $54.10_{\pm 1.4}$ |
| ResNet-34 | $66.10_{\pm 0.25}$ | $54.85_{\pm 0.75}$ |
| ResNet-50 | $66.25_{\pm 0.22}$ | $55.11_{\pm 0.52}$ |

to large, by adjusting the exemplar size for all methods while using a single benchmark backbone, as in Mai et al. (2021). Fig. 11 illustrates the performance-memory curve for CIFAR-100, while Table 8 provides the AUC values for average performance-memory (AUC-A). Both Fig. 11 and Table 8 reveal that beyond a memory size of 44 MB (when the number of generated images exceeds $1k$), the performance of our method stabilizes. This demonstrates that our method is more scalable and extendable compared to other evaluated.

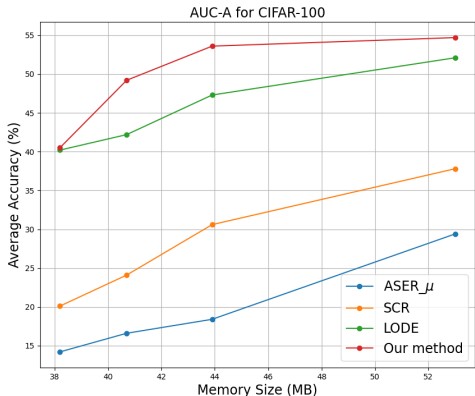

Figure 11: The average accuracy of various methods is evaluated over a range of memory sizes, from small to large. All three compared exemplar-based methods utilize a benchmark backbone, as in Mai et al. (2021). To ensure fair comparison, memory costs are aligned by adding exemplars to the compared methods. For our method, the number of generated images is adjusted to match different memory size requirements.

Table 8: Memory-agnostic performance measures (AUC-A)

| Schemes | CIFAR-100 |
|---|---|
| | $K = 10$ |
| ASER$_\mu$ Shim et al. (2021) | 311.9 |
| SCR Mai et al. (2021) | 453.9 |
| LODE Liang & Li (2024) | 698.5 |
| Our method | **760.3** |

## 5 Conclusion

In this paper, we ventured into the realm of class-incremental learning (class-IL), utilizing the potent tool of metric learning. Instead of storing any actual past data samples, we stored and re-used a single past-trained encoder in a recursive manner to generate the past synthetic images. Leveraging these synthesized images, we introduced three novel rules, or metric KD regularizers, designed to mitigate the issue of catastrophic forgetting, where previous knowledge is lost when learning new classes. Additionally, we incorporated an inter-task confusion regularizer to alleviate the overlap among distinct classes within the embedding space. Experimental results showed the superior performance of our approach compared to many recent approaches.

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

## A  Appendix

### A.1  Choosing Hyper-parameters

In this subsection, we examine the choice of the number of synthetic images and the values for the loss term coefficients in equations (1) and (3) across different simulation setups. Fig. 12 shows the final accuracy of our method on CIFAR-100 using various amounts of synthetic images. It is evident that beyond one-fourth of the real dataset, performance improvement becomes negligible. Therefore, we set the number of synthetic images to one-fourth of the real data for each task across all simulation settings.

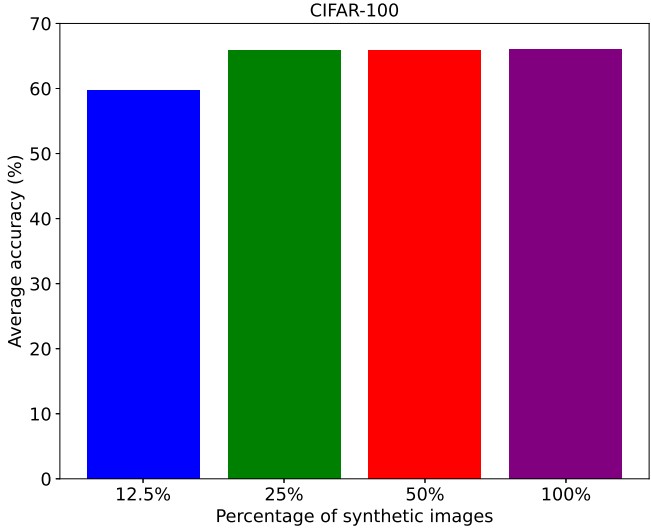

Figure 12: The effect of different numbers of synthetic images on the final average for the CIFAR-100 dataset.

Table 9 presents the results of performance evaluations across various loss term coefficient values in (1) and (3). The determination of final coefficients follows a specific approach. Through empirical observations, it was identified that employing significantly distinct values for $\lambda_1$ and $\lambda_2$, or setting $\lambda_1 = \lambda_2$ with $\lambda_1 > \lambda_3$, results in poor accuracy. Conversely, opting for $\lambda_1 = \lambda_2$ and selecting $\lambda_3 > \lambda_1$ yields improved performance, and the specific value assigned to $\lambda_1$ has minimal impact on the ultimate classification accuracy, as long as it satisfies the constraint $0 < \lambda_1 < 1$.

Table 9: Impact of hyper-parameters in (1) and (3) on average accuracy, $A_K$ (%).

| Schemes | TinyImageNet | | | | CIFAR-100 | | | |
|---|---|---|---|---|---|---|---|---|
| | $\lambda_1$ | $\lambda_2$ | $\lambda_3$ | $A_K$ | $\lambda_1$ | $\lambda_2$ | $\lambda_3$ | $A_K$ |
| | 0.1 | 0.1 | 0.2 | 52.88 | 0.1 | 0.1 | 0.2 | 64.00 |
| | 0.2 | 0.2 | 0.3 | 54.20 | 0.2 | 0.2 | 0.3 | 64.65 |
| | 0.3 | 0.3 | 0.4 | 54.00 | 0.3 | 0.3 | 0.4 | 64.28 |
| | 0.4 | 0.4 | 0.5 | 53.72 | 0.4 | 0.4 | 0.5 | 64.20 |
| Our method | 0.2 | 0.2 | 0.5 | 54.55 | 0.2 | 0.2 | 0.5 | 64.96 |
| | 0.2 | 0.2 | 0.6 | 54.40 | 0.2 | 0.2 | 0.6 | 65.14 |
| | **0.2** | **0.2** | **0.4** | **54.92** | 0.2 | 0.2 | 0.7 | 65.60 |
| | 0.2 | 0.2 | 0.3 | 54.15 | **0.2** | **0.2** | **0.8** | **65.85** |
| | 0.1 | 0.1 | 0.4 | 53.91 | 0.2 | 0.2 | 0.9 | 65.52 |
| | 0.1 | 0.1 | 0.5 | 53.25 | 0.1 | 0.1 | 0.8 | 64.83 |

The hyper-parameter $\lambda_b$ in equation (3) controls the weight of the feature distribution regularization term in DeepInversion. A well-balanced setting ensures effective alignment of feature statistics without excessive

regularization, thereby enhancing generalization and improving model performance. To identify the proper value of $\lambda_b$, we conducted experiments varying it across the range $[5, 10, 15, 20, 50]$. The results, summarized in Table 10, demonstrate that higher-resolution datasets typically require larger values of $\lambda_b$ for better performance.

Table 10: The effect of $\lambda_b$ on Average accuracy, $A_K$ (%).

| $\lambda_b$ | CIFAR-100 | TinyImageNet |
|---|---|---|
| | $K = 10$ | $K = 20$ |
| $\lambda_b = 5$ | $54.6_{\pm 0.3}$ | $53.9_{\pm 1.1}$ |
| $\lambda_b = 10$ | $\mathbf{54.8}_{\pm 0.9}$ | $53.8_{\pm 0.9}$ |
| $\lambda_b = 15$ | $54.5_{\pm 0.6}$ | $\mathbf{54.1}_{\pm 1.4}$ |
| $\lambda_b = 20$ | $54.1_{\pm 0.7}$ | $53.9_{\pm 1.2}$ |
| $\lambda_b = 50$ | $53.6_{\pm 0.6}$ | $53.3_{\pm 1.4}$ |

## A.2 Computation Efficiency Analysis

The additional component of our method, compared to baseline approaches without data generation (e.g., fine-tuning, LwF, and EWC), is the generation of synthetic samples during training. Hence, to evaluate the computational efficiency of our approach, we measured the time required to generate a batch of 16 images across different image sizes using a single NVIDIA V100 GPU. The experiments were conducted over 10 independent trials, with each trial running 100 iterations. The average time per batch was recorded using the Python `time` module, and the results are summarized in Table 11. Even for high-resolution images ($224 \times 224 \times 3$), the generation process demonstrated acceptable computational efficiency, requiring only 3.38 seconds per batch. This indicates that the training complexity, even for large image sizes, is not a significant issue in practice.

Table 11: Computation efficiency for generating a batch of 16 images with 100 iterations on a single NVIDIA V100 GPU.

| Image Size | Mean Time per Batch (seconds) |
|---|---|
| $32 \times 32 \times 3$ | 2.06 |
| $64 \times 64 \times 3$ | 2.11 |
| $84 \times 84 \times 3$ | 2.34 |
| $224 \times 224 \times 3$ | 3.38 |

## A.3 Noisy Triplet Loss

The term $L_{tri}$ in (1) is a triplet loss used to learn the current task. In this paper, we use a very simple, yet effective way to construct the triplet loss for mitigating over-fitting while learning the current task. In general, when the model learns how to make the distances between anchor-positive pairs smaller than those of anchor-negative pairs, the learning process will stop; however, this does not necessarily prevent the model from being over-fitted. In our proposed scheme, as a regularization method, we add noise to some of the embeddings in order to alleviate over-fitting. To be more precise, let $\tilde{H}_k = [\tilde{h}_k^1, \ldots, \tilde{h}_k^b] \in \mathbb{R}^{p \times b}$ denote the un-normalized embedding matrix produced by $M_k$ and let $H_k = [h_k^1, \ldots, h_k^b] \in \mathbb{R}^{p \times b}$ denote its normalized version with $h_k^j = \frac{\tilde{h}_k^j}{\|\tilde{h}_k^j\|}$ where $\|.\|$ denotes $l_2$ norm. We add noise to some normalized embeddings $h_k^j$ so that the pairwise distances for triplets become randomly different in each mini-batch. Specifically, we randomly select a subset of the normalized embeddings in each mini-batch. Then we add Gaussian noise to the selected $h_k^j$, followed by creating triplets. This helps the network to learn the features with a lower risk of over-fitting while fulfilling the triplet loss inequality.

For generating noisy triplets, as the added noise, we choose zero-mean Gaussian noise $w$ with variance $\sigma^2$, i.e., $w \sim \mathcal{N}(0, \sigma^2)$. The noise variance, $\sigma^2$, is an essential hyper-parameter. Based on our experiments, we

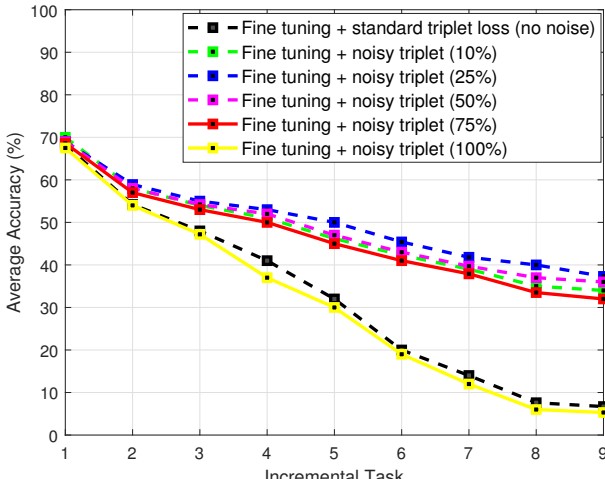

Figure 13: Average accuracy obtained by fine-tuning ResNet-18 using the classical triplet loss and noisy triplet loss with five different portions of noisy triplets in each mini-batch. The model is trained on CIFAR-100 (best viewed in color). NCM is used in the test time, the results indicate a significant improvement in the final task (about 30%) using noisy triplet loss.

find that it should be less than 0.01 for all datasets. In the simulations, we set $\sigma^2 = 0.005$ for CIFAR-10 and $\sigma^2 = 0.01$ for all three CIFAR-100, Mini-ImageNet, and TinyImageNet.

One interesting (and important) question is whether the noisy triplet loss is still effective for general IL (i.e., other than our proposed IL method). To answer the question, we compare the noisy triplet loss and the standard (traditional) triplet loss in the baseline IL setting in which vanilla fine-tuning is used in each task. In the simulation, we train a single encoder, a ResNet-18, on CIFAR-100. The squared Euclidean distance is utilized as the distance metric in the triplet loss. The performance of the traditional triplet loss and noisy triplet loss is compared in Fig. 13, in which the average accuracy is plotted for the case of no noise and for the case of adding noise to 10% of embeddings, 25% of embeddings, 50% of embeddings, 75% of embeddings, and 100% of embeddings. The effectiveness of noisy triplet loss is clear from Fig. 13, and the best performance is achieved when the noise is added to 25% of the embeddings. The accuracy improvement is about 30% at the end of the training (i.e., at the incremental task of 9).

