# OpenReview forum: "Combating Inter-Task Confusion and Catastrophic Forgetting by Metric Learning and Re-Using a Past Trained Model"
_TMLR — Accepted by TMLR_

### Review · Reviewer_RT7T · 2024-12-31

**Summary Of Contributions:**

This paper proposes a new algorithm to alleviate catastrophic forgetting in Class-IL.The authors first explain that there are multiple sources for the poor performance of the models in incremental learning. First, the training data of prior tasks are not available during the training of the future tasks; second, the model has not learned to distinguish between classes from different tasks.

They propose a combination of loss terms to train the model with two goals in mind. First, remind the model of the past task and help it to identify classes from different tasks.

**Audience:**

Yes

**Claims And Evidence:**

Yes

**Requested Changes:**

* Some methods and losses are mentioned very early and never discussed properly or introduced very late. For example, metric learning is briefly mentioned in section 2.2, and it plays an essential role in the final algorithm but is not adequately explained. The same goes for *l*$^{mse}$, defined two pages after introducing it. Another example is $h_k$, shown in a figure on page 2 and used on page 4 without any description.

* In section 3.1, only average accuracy is mentioned as the evaluation metric, and the forgetting is missing.

* Can you explain the eq 2 (L$_{TCR}$)? What is positive input ($x_p$)? Are the inputs in embedding space? What is the learnable parameter here?

* The author's contributions are not very clear. It would be helpful to point out the terms directly brought from previous literature in continual learning versus the losses that the authors propose. For example, is $l_{mse}$ a new loss? It looks pretty similar to equation 12 in [1].


[1] Smith, James, et al. "Always be dreaming: A new approach for data-free class-incremental learning." Proceedings of the IEEE/CVF international conference on computer vision. 2021.

**Strengths And Weaknesses:**

**Strengths**

1- The authors mention some exciting findings. For example, the best synthetic data for task i can be generated by the model trained on task i. So, it is better to train the generative model, generate the synthetic data, and use them for future training.

2- The authors show superior performance comparing the prior works.

3- The ablation study is comprehensive.

**Weakness**

1- The paper is tough to follow and lacks organization. Many parameters are employed in different components of the algorithms, but some of the methods are not properly referenced or defined. Also, a diagram or algorithm block could help the reader understand each part and the overall algorithm.

2- The authors' novelty is not very clear.

3- Please check out the requested changes.

---

> ### Author Response · Authors · 2025-01-23
> **We sincerely thank the reviewer for the valuable comments and suggestions. We have revised the paper based on these comments and uploaded the updated manuscript with changes highlighted in yellow.**
>
> $\textbf{Comment 1}$: Some methods and losses are mentioned very early and never discussed properly or introduced very late. For example, metric learning is briefly mentioned in section 2.2, and it plays an essential role in the final algorithm but is not adequately explained. The same goes for $l^{mse}$, defined two pages after introducing it. Another example is $h_k$, shown in a figure on page 2 and used on page 4 without any description.
>
> $\textbf{Response 1}$: Thank you for the thoughtful comment. To address this comment, we have added several paragraphs to provide a clearer discussion on metric learning, $L_{KD}$, and $H_k$.
>
> To have a clearer explanation on metric learning and its advantages compared to the softmax classifier, we added two paragraphs and two figures (Fig. 2 and Fig. 3) into Section 2.2.
>
>
> To have a better discussion on $L_{KD}^{mse}$, $L_{KD}^{intra}$, and $L_{KD}^{inter}$ right after introducing them, we have extended Section 3.4 as follows:
>
> “$$ L_{KD}=\lambda_2 L_{KD}^{mse}+\lambda_3 (L_{KD}^{intra}+L_{KD}^{inter}), $$
>
> where $\lambda_2$ and $\lambda_3$ are hyper-parameters to control the impact of each term. The first term $L_{KD}^{mse}$ aligns embeddings of the current and past models to ensure that the current model retains knowledge of past tasks. To that end we minimize the Mean Square Euclidean (MSE) distance between the two sets of embeddings produced by current trainable model $M_k$ and the stored (fixed) last model $M_{(k-1)}$.
>
> The term $L_{KD}^{intra}$ preserves the embedding consistency within each class across tasks and the term $L_{KD}^{inter}$ maintains the relative positions and separations between different classes in the embedding space. To construct these three KD losses, we utilize the synthetic data $G_k$ of the last task as inputs to both $M_k$ and $M_{(k-1)}$. Therefore, we will begin by detailing the process of generating $G_k$ in the following subsection. Subsequently, each of the three KD losses will be explained one by one.”
>
>
> To have a better explanation on $H_k$, we extended end of first paragraph of Section 3.2 as follows:
>
> “In our proposed method, the current model $M_k$ is trained by $L_{tri}$  to learn the current task using the embeddings $H_k$ which are produced when the dataset $D_k$ of the actual data samples for the current task are supplied as input. To be more precise $H_k=[h_k^1,…,h_k^b ]∈R^{(p×b)}$ denote the normalized embedding matrix produced by trainable student model $M_k$ using real current task dataset. The dimension of embedding space is $p$ and batch size is $b$.”
>
> $\textbf{Comment 2}$: In section 3.1, only average accuracy is mentioned as the evaluation metric, and the forgetting is missing.
>
> $\textbf{Response 2}$: Thank you for your valuable comment. To address this comment, we have moved the definition of forgetting to Section 3.1.
>
> $\textbf{Comment 3}$: Can you explain the eq 2 ($L_{TCR}$)? What is positive input ($x_p$)? Are the inputs in embedding space? What is the learnable parameter here?
>
> $\textbf{Q1}$: Can you explain the eq 2 ($L_{TCR}$)?
>
> $\textbf{Response Q1}$: The Inter-Task Confusion Regularizer ($L_{TCR}$) specifically uses the centroids of previously learned classes in the embedding space as negatives and randomly selected data points (in the embedding space) in every class of the current task as positives. In other words, the positives are a random subset of embeddings $H_k$ which are produced when the dataset $D_k$ of the actual data samples for the current task are supplied as input to trainable model $M_k$.
>
> Our $L_{TCR}$ loss inherently addresses the issue of task confusion by encouraging the distance between current class embeddings and past class centroids to be maximized. This approach not only reduces overlaps between classes from different tasks, as shown in Fig. 4(c), but also effectively mitigates the impact of task confusion. By maximizing these distances, $L_{TCR}$ creates a buffer in the embedding space, ensuring that even if distribution shifts occur (in an unknown direction), the separation between classes remains robust.
>
> This iterative process ensures that each task is incrementally learned such that:
>
> 1- The embeddings of the current class are separated from all previously learned classes.
>
> 2- The separation and structure established during the learning of a class are preserved when subsequent tasks are introduced
>
> $\textbf{Q2}$: What is positive input ($x_p$)?
>
> $\textbf{Response Q2}$: The positives are a random subset of embeddings $H_k$ which are produced when the dataset $D_k$ of the actual data samples for the current task are supplied as input to trainable model $M_k$.
>
> $\textbf{Q3}$: Are the inputs in embedding space?
>
> $\textbf{Response Q3}$: Yes, the $L_{TCR}$ totally works in the embedding space.
>
> $\textbf{Q4}$: What is the learnable parameter here?
>
> $\textbf{Response Q4}$: The learnable parameters are the parameters of learnable model $M_k$, since positives are produced by $M_k$.

---

> > ### Author Response · Authors · 2025-01-23
> > **Response to last comment**
> >
> > $\textbf{Comment 4}$: The author's contributions are not very clear. It would be helpful to point out the terms directly brought from previous literature in continual learning versus the losses that the authors propose. For example, is  a new loss? It looks pretty similar to equation 12 in [1].
> >
> >
> > $\textbf{Response 4}$: Thank you for your comment regarding the contributions of our work. Below, we summarize the unique design elements and contributions of our method that distinguish our work from existing research:
> >
> > 1-Novel Regularization Term for Inter-Task Confusion:
> > - We propose the Inter-Task Confusion Regularizer ($L_{TCR}$), which specifically uses the centroids of previously learned classes as negatives to minimize overlaps between classes from different tasks in the embedding space. While existing methods address catastrophic forgetting, the explicit focus on reducing inter-task confusion by leveraging embedding centroids as part of a customized loss function is a significant innovation.
> > - Unlike traditional methods, $L_{TCR}$ introduces a systematic way to maintain clear class boundaries across tasks, which has not been explicitly tackled in prior work.
> > - To the best of our knowledge, we are the first who proposed this effective mechanism to directly address the inter-task confusion.
> >
> > 2-Dual-Functionality of the Stored Model:
> > - Another novel element of this method is its use of the stored model from the previous task $M_{(k-1)}$ to generate synthetic images and serve as a teacher model for knowledge distillation.
> > - This dual-use approach is unique because most methods that rely on knowledge distillation require additional storage for data and a separate generator model (e.g., VAEs, GANs). By contrast, this paper introduces a streamlined process in which the stored model is repurposed to synthesize new data while also acting as a source of knowledge for the current task model ($M_k$).
> > - This design enables the approach to be both computationally efficient and memory-saving, which is particularly scalable to class-IL settings with extensive tasks.
> > - To the best of our knowledge, we are the first who utilize the store model for two different purposes: generating synthetic images and serving as a teacher model for knowledge distillation.
> >
> > 3-Three Novel Knowledge Distillation Losses in the Embedding Space:
> > - To address catastrophic forgetting, we introduce three KD losses that are complementing with one another:
> > 	+ MSE Loss: Aligns embeddings of the current and past models to ensure that the current model retains knowledge of past tasks.
> > 	+ Intra-Class Regularization: Preserves the embedding consistency within each class across tasks.
> > 	+ Inter-Class Regularization: Maintains the relative positions and separations between different classes in the embedding space.
> > - These losses go beyond standard KD methods by focusing on embedding-level consistency and structure preservation, which are critical for class-incremental learning.
> > - To the best of our knowledge, we are the first who proposed this set of losses in the embedding space to tackle catastrophic forgetting and inter-task confusion.
> >
> > In summary, building upon fundamental techniques such as KD and exemplar-free learning, we have introduced multiple innovative elements that significantly advance the state of the art. These contributions, combined with our practical and scalable design, set our approach apart from existing methods. We believe that the novelty and impact of our work are well-supported by the experimental results and methodological innovations presented in the paper.
> >
> > Comparing our $L_{KD}^{mse}$ with Eq. (12) in [1], we observe that while the two methods are similar, they are not identical. In our approach, we exclusively use synthetically generated images from past tasks to produce embeddings for $L_{KD}^{mse}$. In contrast, [1] incorporates both real and synthetic images to generate embeddings for Eq. (12). Our primary focus for literature review has been on continual learning with metric learning as the backbone, whereas the method in [1] utilizes distinct fully connected layers as separate heads for individual tasks.

---

### Review · Reviewer_rxXC · 2025-01-06

**Summary Of Contributions:**

This work leverages several techniques like Generative Data Replay and Knowledge Distillation to tackle the challenges of class Incremental learning within the metric learning framework. The work claims to tackle two key challenges: inter-task confusion and catastrophic forgetting. To mitigate these issues, they introduce a novel loss function, $\mathcal{L}_{M_k}$ (Equation 1), comprising three components given below

- **$\mathcal{L}_{tri}$** loss s Schroff et al. (2015), is used to train the $k^{th}$ task as it is superior in the IL context Yu et al. (2020).
- **$\mathcal{L}_{TCR}$**  (Equation 2), the inter-task confusion regularizer, specifically aims to prevent confusion between the current task and previously learned tasks.
- The authors propose storing the model trained on the preceding k−1 tasks to combat catastrophic forgetting. This stored model is then used to generate synthetic images of the $(k−1)^{th}$ task. These synthetic images are incorporated during the training of the $k^{th}$ task, ensuring the preservation of the embedding space for previous tasks while simultaneously learning new embeddings (facilitated by the knowledge distillation loss, **$\mathcal{L}_{KD}$** (Equation 3).

The authors claim that their approach is effective by outperforming several baseline methods. Furthermore, they conduct ablation studies to analyze the contributions of each proposed loss term.

**Audience:**

Yes

**Claims And Evidence:**

No

**Requested Changes:**

# Requested changes based on the questions and comments above:
1. **Claim regarding inter-task confusion** [Motivation]
    - Authors state (Page 2, para 2 first line) "many researchers have proposed various approaches to tackle catastrophic forgetting while very few have tackled inter-task confusion explicitly." I would request the reviewers to add citations to a few works that tackle 'inter-task confusion' explicitly.
    - As the authors admit that the concepts of catastrophic forgetting and inter-task confusion have not been well distinguished in literature, I request the authors to add clear definitions that can lead to some measurable metric to test what fraction of performance drop is caused due to each of these concepts.
    - I recommend the authors use these definitions to quantitatively assign what portion of performance improvement is alleviating catastrophic forgetting and inter-task confusion for at least a few of the methods in Table 1.

2. **Generation of only 1 task** [Method]

    - Authors should consider clearly stating that they **only generate sample for 1 previous task**, where the claims regarding the use of generative replay are made. I was under the impression that author were generating the samples for each task until I reached the Section 3.4 (on pg 7).
        - the abstract - "we also propose that the past trained model is stored and re-used for generating past data samples. ",
        -  the introduction - "We first use Mk−1 to synthesize past data samples Gk, which will be used for training Mk." and
        - contribution - "To this end, we also propose a method for synthesizing past data samples by storing and re-using a past trained model Mk−1."
    - **Task scalability**- It would be interesting to see how this algorithm behaves when there are large number of tasks, for example Cifar100 with 20-25 task splits. Will this method be effective against the data replay baselines as this algorithm only generates data for past 1 task?
    - Additionally, authors can consider demonstrating how the embeddings for the first task change with incremental learning.

3. **Comparisons against several works** [Evaluation]
    - I would strongly recommend the authors reimplement a few of the recent baselines for a controlled experiment. Where each of the algorithms gets the same task split and all the algorithms have the same initialization after the first task (as the incremental nature of the algorithm is only apparent from task2). Alternatively ( and probably more simpler approach), the authors can report the number from some other papers containing several methods like [1], provided they able to (a) test the reproducibility of the results reported in the paper and (b) are able to port their method into these repositories and get the numbers corresponding to their method.
    - I would recommend the authors to add a few more architectures in the comparison.
    - I request the authors to add data entry for vanilla training of the network on the entire dataset as the upper bound in all the tables to show how much performance gap comes from the incremental nature of the task.
    - Authors primarily use Average Accuracy to compare several algorithms. The authors should also consider adding several other metrics such as Last model accuracy and backward transfer [2].

4. **Budget normalised comparisons** [Evaluation]
    - The author should report the memory and compute requirements of each of the methods in the Table.
    - Additionally, the authors can use the metrics AUC-L  and AUC-A to compare memory agnostic measures proposed in [1].
    - It can be a good contribution if authors can come up with a compute agnostic measure as the proposed method might requires additional compute to generate the images. (May be this compute is negligible in comparison to other methods, in which case the proposed algorithm would show impressive numbers.)



# Reference
[1] Zhou, Da-Wei, et al. "Class-incremental learning: A survey." IEEE Transactions on Pattern Analysis and Machine Intelligence (2024).

[2] Saha, Gobinda, and Kaushik Roy. "Continual learning with scaled gradient projection." Proceedings of the AAAI Conference on Artificial Intelligence. Vol. 37. No. 8. 2023.

**Strengths And Weaknesses:**

# Strengths
- The paper tackles an important and challenging real-world problem.
- The paper is written well and is easy to follow. I particularly like the illustration presented in Figure 2 as it explains the core intuition behind the proposed loss function.
- This work is compared against several baselines in Table 1 across several small and mid-sized datasets.
- The work analyzes the effect of different loss terms through their ablation study, which shows the significance of each loss term.

# Questions and Comments
1. **Claim regarding inter-task confusion** [Motivation]

    I find the distinction between inter-task confusion and catastrophic forgetting somewhat unclear. Authors have stated (page 2 1st para last line) - "Although these two concepts often have not been clearly distinguished in the literature, the inter-task confusion and the catastrophic forgetting should be treated separately because the former refers to the confusion between the current and past tasks, whereas the latter refers to forgetting the past tasks." As per my understanding, Catastrophic Forgetting occurs due to the new task overwriting some of the information learned previously. In class incremental learning, since the task split is based on classes, learning new classes can inadvertently overwrite information about old classes. This implies that class confusion is the major contributing reason for catastrophic forgetting. Many existing CIL methods mentioned by the authors, such as data replay, regularization, parameter isolation, and hybrid approaches, effectively address catastrophic forgetting by primarily focusing on mitigating inter-task (class) confusion. For instance, naive data replay, which stores samples from past tasks, combats catastrophic forgetting by allowing the model to learn from past data. This simultaneously helps the model distinguish between past and current classes, thereby alleviating the challenge of class confusion.

2. **Generation of only 1 task** [Method]

    Authors use the model trained on $(k-1)^{th}$ task to generate the images only for the previous task. Only these images are used for KL divergence-based loss terms. In my understanding, this should affect the algorithm's ability when the number of tasks is large. Or the task split is designed in a way that the confusing classes are kept far apart.

3. **Accuracy Comparisons against several works** [Evaluation]

    Authors state that the ordering of the classes in crucial for Class Incremental learning (on Page 13). In my experience the final converged model is also sensitive to the initialization, the training setup and the hyperparameters. The authors have made efforts to add several comparison points in Table 1. I believe the authors report the numbers from the original paper for many of the work and this makes the comparison unfair as the different papers would have different task splits and training setups.

4. **Budget normalised comparisons** [Evaluation]

    It might be unfair to compare the performance of the methods as these methods might have different computing and memory requirements. The authors should consider adding a discussion on the budget requirement of different methods.

---

> ### Author Response · Authors · 2025-01-23
> **Responses to the rxXC Reviewers’ Comments**
>
> We sincerely thank the reviewer for the valuable comments and suggestions. We have revised the paper based on these comments and uploaded the updated manuscript with changes highlighted in yellow.
>
> $\textbf{Comment 1}$:  Claim regarding inter-task confusion
>
> $\textbf{a)}$ Authors state (Page 2, para 2 first line) "many researchers have proposed various approaches to tackle catastrophic forgetting while very few have tackled inter-task confusion explicitly." I would request the reviewers to add citations to a few works that tackle 'inter-task confusion' explicitly.
>
> $\textbf{Response 1(a)}$: Thank you for the thoughtful comment. Following works tackle inter-task confusion explicitly and we added those into our references.
>
> [Ref.1] Nori, M. K., & KIM, I. M. (2024). Task Confusion and Catastrophic Forgetting in Class-Incremental Learning: A Mathematical Framework for Discriminative and Generative Modelings. In The Thirty-eighth Annual Conference on Neural Information Processing Systems.
>
> [Ref.2] Huang, B., Chen, Z., Zhou, P., Chen, J., & Wu, Z. (2023, June). Resolving task confusion in dynamic expansion architectures for class incremental learning. In Proceedings of the AAAI Conference on Artificial Intelligence (Vol. 37, No. 1, pp. 908-916).
>
> $\textbf{b)}$ As the authors admit that the concepts of catastrophic forgetting and inter-task confusion have not been well distinguished in literature, I request the authors to add clear definitions that can lead to some measurable metric to test what fraction of performance drop is caused due to each of these concepts.
>
> $\textbf{Response 1(b)}$: Thank you for the thoughtful comment. Here is the inter-task confusion definition:
>
> Inter-task confusion (ITC) arises in class-incremental learning (class-IL) when a model struggles to differentiate between classes belonging to different tasks during testing. This occurs because the classes from distinct tasks are never seen together during training, and the model is not explicitly taught to distinguish between them. As a result, the learned features are not optimized to discriminate across tasks, leading to confusion when all tasks are evaluated together without access to task-specific identifiers.
>
> For instance, while each task's classes are trained separately to distinguish their own categories, the discrepancy between classes from different tasks remains unresolved, resulting in misclassifications across tasks. This phenomenon becomes particularly evident when task-IDs must be inferred at test time.
>
> On the other hand, catastrophic forgetting (CF) refers to the phenomenon in which a model experiences a significant drop in performance on tasks it previously learned after learning a new task. This issue is indicative of the model either entirely or substantially forgetting what it had already learned.
>
> $\textbf{c)}$ I recommend the authors use these definitions to quantitatively assign what portion of performance improvement is alleviating catastrophic forgetting and inter-task confusion for at least a few of the methods in Table 1.
>
> $\textbf{Response 1(c)}$: Thank you for the thoughtful comment. To address this comment, we can quantify the portion of inter-task confusion ($ITC$) based on following metric:
>
> $$ITC = (Accuracy given task-IDs) - (Accuracy without task-IDs),$$
>
> We have added Fig. 10 to Subsection 4.5 to show the portions of catastrophic forgetting (CF) and inter-task confusion (ITC) for fine tuning and three of recent methods including ours.
>
> $\textbf{Comment 2}$: Generation of only 1 task [Method]
>
> $\textbf{a)}$ Authors should consider clearly stating that they only generate sample for 1 previous task, where the claims regarding the use of generative replay are made. I was under the impression that author were generating the samples for each task until I reached the Section 3.4 (on pg 7).
> - the abstract - "we also propose that the past trained model is stored and re-used for generating past data samples. ",
> - the introduction - "We first use $M_{k−1}$ to synthesize past data samples $G_k$, which will be used for training $M_k$." and
> - contribution - "To this end, we also propose a method for synthesizing past data samples by storing and re-using a past trained model $M_{k−1}$."
>
> $\textbf{Response 2(a)}$:  Thank you for the thoughtful comment. To address this comment, we explicitly mentioned that we only generate samples only for the very last task in all parts as follow:
>
> - the abstract - "we also propose that the past trained model is stored and re-used for generating past data samples for only one previous task. ",
> - the introduction - "We first use $M_{k−1}$ to synthesize past data samples $G_k$ belong to the very last task, which will be used for training $M_k$." and
> - contribution - "To this end, we also propose a method for synthesizing past data samples by storing and re-using a past trained model $M_{k−1}$. Using $M_{k−1}$, at current task k, we only generate samples belong to $(k-1)$-th task."

---

> > ### Author Response · Authors · 2025-01-23
> > **Responses to the rxXC Reviewers’ Comments**
> >
> > $\textbf{b)}$ Task scalability- It would be interesting to see how this algorithm behaves when there are large number of tasks, for example Cifar100 with 20-25 task splits. Will this method be effective against the data replay baselines as this algorithm only generates data for past 1 task? Additionally, authors can consider demonstrating how the embeddings for the first task change with incremental learning.
> >
> > $\textbf{Response 2(b)}$:  Thank you for the thoughtful comment. To address the comment, about number of tasks for CIFAR-100 and TinyImageNet we provided results considering 20 tasks split in Table 2 and final last accuracy in Table 3.
> >
> > **Table 2.** Average accuracy, $A_K$ (%), after finishing the final task setting half of the classes for the first task.
> >
> > | **Schemes**         | **CIFAR-100 (K=5)** | **CIFAR-100 (K=10)** | **CIFAR-100 (K=20)** | **TinyImageNet (K=5)** | **TinyImageNet (K=10)** | **TinyImageNet (K=20)** |
> > |----------------------|---------------------|----------------------|----------------------|------------------------|-------------------------|-------------------------|
> > | Joint training | 74.32              | 74.32               | 74.32               | 66.81                 | 66.81                  | 66.81                  |
> > | SDC | 56.77      | 57.00        | 58.90        | ★                     | ★                      | ★                      |
> > | Pass | 63.47       | 61.84        | 58.09        | 49.55          | 47.29           | 42.07           |
> > | S-SRE | 65.88       | 65.04        | 61.70        | 50.39          | 48.93           | 48.17          |
> > | LODE | 65.88     | 65.04        | 61.70        | 50.39         | 48.93          | 48.17         |
> > | **Our method**  | **66.15 ± 0.10**   | **65.82 ± 0.30**    | **64.35 ± 0.70**    | **55.32 ± 1.2**       | **54.90 ± 1.5**        | **54.10 ± 1.4**        |
> >
> > **Table 3.** Final task accuracy, $A_{last}$ (%).
> >
> > | **Schemes**        | **CIFAR-100 (K=5)** | **CIFAR-100 (K=10)** | **CIFAR-100 (K=20)** | **TinyImageNet (K=5)** | **TinyImageNet (K=10)** | **TinyImageNet (K=20)** |
> > |---------------------|---------------------|----------------------|----------------------|------------------------|-------------------------|-------------------------|
> > | Joint training      | 74.32              | 74.32               | 74.32               | 66.81                 | 66.81                  | 66.81                  |
> > | SCR                | 31.22              | 28.65               | 26.10               | 29.15                 | 27.45                  | 23.11                  |
> > | ESMER              | 49.90              | 48.77               | 47.35               | 48.85                 | **47.30**              | 48.40                  |
> > | IMEX-Reg           | 50.10              | 48.54               | 46.25               | 49.65                 | 46.64                  | 41.34                  |
> > | LODE               | 51.00              | 46.31               | 46.00               | 50.20                 | 46.80                  | **45.11**              |
> > | **Our method**     | **51.25**          | **49.67**           | **47.83**           | **50.76**             | 47.13                  | 44.85                  |
> >
> > $\textbf{Comment 3}$: Comparisons against several works [Evaluation]
> >
> > $\textbf{a)}$ I would strongly recommend the authors reimplement a few of the recent baselines for a controlled experiment. Where each of the algorithms gets the same task split and all the algorithms have the same initialization after the first task (as the incremental nature of the algorithm is only apparent from task2). Alternatively ( and probably more simpler approach), the authors can report the number from some other papers containing several methods like [1], provided they able to (a) test the reproducibility of the results reported in the paper and (b) are able to port their method into these repositories and get the numbers corresponding to their method.
> >
> > $\textbf{Response 3(a)}$: Thank you for the constructive comment. To address this comment, we employed the repository available at https://github.com/RaptorMai/online-continual-learning to replicate the results for the baseline methods: EWC++, LwF, AGEM, ER, GSS, MIR, ASER_μ, and SCR, within a controlled experimental framework. In all experiments, the data splits and the random seeds used for class selection were consistent across trials, with the reported results representing the average of 10 different seeds (task orderings). Additionally, we reproduced the average accuracy shown in Table 1 for the methods ESMER, IMEX-Reg, CW-DPPER, and LODE, using the same data splits and task orderings. However, for the methods Gen., CF-IL, Semi, and MGI, we were unable to replicate the results based on the available source code; consequently, we extracted the results directly from the respective original papers.

---

> > > ### Author Response · Authors · 2025-01-23
> > > **Responses to the rxXC Reviewers’ Comments**
> > >
> > > $\textbf{b)}$ I would recommend the authors to add a few more architectures in the comparison.
> > >
> > > $\textbf{Response 3(b)}$: Thank you for the thoughtful comment. We added results for Resnet-34 and Resnet-50 only for CIFAR-100 and TinyImageNet in Table.7 of Section 4.5. We can see that the improvement by larger models is not noticeable for CIFAR-100 but for TinyImageNet the improvement is clear.
> > >
> > > ### Table 7. Average accuracy, $A_K$ (%) for different networks.
> > >
> > > | **Architecture** | **CIFAR-100 (K=10)** | **TinyImageNet (K=20)** |
> > > |-------------------|-----------------------|-------------------------|
> > > | ResNet-18         | 65.82 ± 0.3          | 54.10 ± 1.4            |
> > > | ResNet-34         | 66.10 ± 0.25         | 54.85 ± 0.75           |
> > > | ResNet-50         | 66.25 ± 0.22         | 55.11 ± 0.52           |
> > >
> > > $\textbf{c)}$ I request the authors to add data entry for vanilla training of the network on the entire dataset as the upper bound in all the tables to show how much performance gap comes from the incremental nature of the task.
> > >
> > > $\textbf{Response 3(c)}$: Thank you for the thoughtful comment. To address this comment, we have added joint training as upper bound in all tables, so we can see the gap between methods and upper bound.
> > >
> > > $\textbf{d)}$ Authors primarily use Average Accuracy to compare several algorithms. The authors should also consider adding several other metrics such as Last model accuracy and backward transfer [2].
> > >
> > > $\textbf{Response 3(d)}$: Thank you for the thoughtful comment. To address this comment, we have provided last model accuracy in Table 3 (provided in Response 2(b)) considering four most recent methods and comparing with our method. We have also provided forgetting (negative BWT) after each incremental task in Table 4. We hope these metrics provide satisfactory performance evolutions.
> > >
> > > **Table 4.** Forgetting (%) and average accuracy (%) after each task for our method, LODE, and the lower bound (i.e., fine-tuning).
> > >
> > > ### CIFAR-10
> > >
> > > | **Method**    | **Metric**    | **k=1** | **k=2** | **k=3** | **k=4** | -     | -     | -     | -     | -     |
> > > |---------------|---------------|---------|---------|---------|---------|-------|-------|-------|-------|-------|
> > > | Fine tuning   | Forgetting    | 9.5     | 12.2    | 15.7    | 28.6    | -     | -     | -     | -     | -     |
> > > |               | Accuracy      | 50.9    | 40.0    | 30.0    | 16.9    | -     | -     | -     | -     | -     |
> > > | LODE          | Forgetting    | 4.2     | 8.4     | 5.1     | 9.4     | -     | -     | -     | -     | -     |
> > > |               | Accuracy      | 90.5    | 81.9    | 79.8    | 75.4    | -     | -     | -     | -     | -     |
> > > | Our method    | Forgetting    | 4.6     | 4.7     | 5.4     | 11.1    | -     | -     | -     | -     | -     |
> > > |               | Accuracy      | 88.1    | 83.3    | 81.1    | 75.9    | -     | -     | -     | -     | -     |
> > >
> > > ---
> > >
> > > ### CIFAR-100
> > >
> > > | **Method**    | **Metric**    | **k=1** | **k=2** | **k=3** | **k=4** | **k=5** | **k=6** | **k=7** | **k=8** | **k=9** |
> > > |---------------|---------------|---------|---------|---------|---------|---------|---------|---------|---------|---------|
> > > | Fine tuning   | Forgetting    | 3.8     | 12.1    | 7.0     | 7.9     | 10.2    | 11.4    | 5.9     | 4.3     | 4.0     |
> > > |               | Accuracy      | 69.8    | 54.4    | 49.1    | 41.8    | 32.2    | 20.0    | 14.6    | 9.1     | 6.5     |
> > > | LODE          | Forgetting    | 5.0     | 10.0    | 9.2     | 2.7     | 1.5     | 1.2     | 1.2     | 1.7     | 9.4     |
> > > |               | Accuracy      | 86.3    | 74.1    | 67.3    | 64.7    | 62.8    | 61.1    | 60.4    | 58.4    | 51.5    |
> > > | Our method    | Forgetting    | 5.3     | 8.6     | 3.2     | 2.9     | 2.6     | 2.2     | 1.0     | 1.7     | 4.1     |
> > > |               | Accuracy      | 82.2    | 71.8    | 69.5    | 66.7    | 63.4    | 60.2    | 59.6    | 57.5    | 54.5    |
> > >
> > > ---
> > >
> > > ### Mini-ImageNet
> > >
> > > | **Method**    | **Metric**    | **k=1** | **k=2** | **k=3** | **k=4** | **k=5** | **k=6** | **k=7** | **k=8** | **k=9** |
> > > |---------------|---------------|---------|---------|---------|---------|---------|---------|---------|---------|---------|
> > > | Fine tuning   | Forgetting    | 12.1    | 18.9    | 14.5    | 5.3     | 9.3     | 4.8     | 5.4     | 3.5     | 2.1     |
> > > |               | Accuracy      | 51.5    | 32.1    | 20.0    | 18.9    | 11.0    | 9.8     | 6.1     | 5.5     | 5.2     |
> > > | LODE          | Forgetting    | 8.7     | 9.7     | 4.8     | 3.4     | 3.0     | 3.1     | 2.0     | 9.7     | 9.1     |
> > > |               | Accuracy      | 79.3    | 70.8    | 64.4    | 62.6    | 59.2    | 56.0    | 54.1    | 50.0    | 48.8    |
> > > | Our method    | Forgetting    | 11.1    | 6.2     | 3.4     | 3.7     | 2.2     | 2.3     | 2.0     | 4.2     | 6.6     |
> > > |               | Accuracy      | 76.8    | 71.4    | 66.8    | 64.8    | 62.3    | 59.9    | 58.0    | 56.2    | 55.3    |

---

> > > > ### Author Response · Authors · 2025-01-23
> > > > **Responses to the rxXC Reviewers’ Comments**
> > > >
> > > > $\textbf{Comment 4}$: Budget normalised comparisons [Evaluation]
> > > >
> > > > $\textbf{a)}$ The author should report the memory and compute requirements of each of the methods in the Table.
> > > >
> > > > $\textbf{Response 4(a)}$: Thank you for the thoughtful comment. To address this comment, we have quantified memory requirements for each method and added to the last three columns of Table 1.
> > > >
> > > > Regarding compute requirements, we have provided consuming time for generating a batch of data in Table 11 of section A.2. Since the additional component of our method, compared to baseline approaches without data generation (e.g., fine-tuning, LwF, and EWC), is the generation of synthetic samples during training, to evaluate the computational efficiency of our approach, we measured the time required to generate a batch of 16 images across different image sizes using a single NVIDIA V100 GPU.
> > > >
> > > > ### Table 1. Average accuracy, $A_K$ (%), after finishing the final task. Our method considerably outperforms all methods, including the coreset replay-based methods on all three datasets. The ‘MS-10,’ ‘MS-100,’ and ‘MS-Mini’ denote the memory size (MB) for CIFAR-10, CIFAR-100, and Mini-ImageNet, respectively.
> > > >
> > > > | **Schemes**      | **MNIST**        | **CIFAR-10**     | **CIFAR-100**    | **Mini-ImageNet** | **MS-10** | **MS-100** | **MS-Mini** |
> > > > |-------------------|------------------|------------------|------------------|-------------------|-----------|------------|-------------|
> > > > | Joint training    | 98.1 ± 0.06     | 83.9 ± 0.6       | 74.3 ± 1.2       | 72.1 ± 0.2        | ★         | ★          | ★           |
> > > > | Fine-tune         | 18.2 ± 0.1      | 16.9 ± 1.1       | 5.4 ± 0.9        | 4.3 ± 0.95        | 42.6      | 42.6       | 42.6        |
> > > > | LwF              | 20.4 ± 0.6      | 20.4 ± 0.6       | 13.5 ± 0.5       | 8.5 ± 0.50        | 42.6      | 42.6       | 42.6        |
> > > > | EWC++            | 61.33 ± 0.10    | 18.2 ± 0.2       | 5.5 ± 0.33       | 4.3 ± 0.30        | 140.4     | 140.6      | 140.6       |
> > > > | AGEM             | 67.9 ± 0.20     | 28.9 ± 1.20      | 14.0 ± 0.45      | 11.5 ± 0.40       | 108.9     | 108.9      | 199.4       |
> > > > | ER               | 66.5 ± 0.35     | 49.1 ± 1.1       | 28.1 ± 1.2       | 20.9 ± 1.5        | 46.9      | 48.4       | 57.4        |
> > > > | GSS              | 71.2 ± 0.23     | 46.4 ± 1.9       | 25.4 ± 0.55      | 20.7 ± 1.5        | 95.1      | 95.1       | 104.2       |
> > > > | MIR              | 71.8 ± 0.30     | 49.3 ± 1.2       | 27.3 ± 1.0       | 21.8 ± 1.0        | 95.1      | 108.9      | 198.4       |
> > > > | ASER$_\mu$       | 75.7 ± 0.40     | 50.2 ± 1.1       | 29.4 ± 0.7       | 21.0 ± 0.43       | 62.1      | 62.1       | 152.6       |
> > > > | SCR              | 78.8 ± 0.25     | 65.6 ± 0.55      | 37.8 ± 0.6       | 35.2 ± 0.55       | 48.3      | 52.9       | 89.1        |
> > > > | Gen.             | 93.79 ± 0.08    | 56.0 ± 0.04      | 49.5 ± 0.06      | ★                 | 99.2      | 99.2       | ★           |
> > > > | CF-IL            | 95.3 ± 0.15     | 75.34            | ★               | ★                 | 59.2      | ★          | ★           |
> > > > | Semi             | 92.3 ± 0.10     | 57.9 ± 1.1       | 38.9 ± 0.5       | ★                 | 96.8      | 102.3      | ★           |
> > > > | MGI              | 90.8 ± 0.30     | 52.1 ± 2.5       | 24.1 ± 0.8       | 19.1 ± 0.9        | 96.2      | 99.8       | 135.4       |
> > > > | ESMER            | 93.5 ± 0.20     | 73.15 ± 0.54     | 50.8 ± 0.31      | ★                 | 94.2      | 94.2       | 97.8        |
> > > > | IMEX-Reg         | 95.7 ± 0.10     | 74.6 ± 0.18      | 50.3 ± 0.23      | ★                 | 98.9      | 98.9       | 102.5       |
> > > > | CW-DPPER         | 91.8 ± 0.20     | 67.42 ± 1.20     | 49.15 ± 0.61     | 51.63 ± 0.11      | 99.7      | 99.7       | 135.9       |
> > > > | LODE             | 95.5 ± 0.90     | 76.3 ± 0.90      | 51.4 ± 1.01      | 52.4 ± 0.13       | 49.7      | 63.9       | 59.7        |
> > > > | **Our method**   | **96.1 ± 0.8**  | **76.9 ± 0.8**   | **54.8 ± 0.9**   | **54.2 ± 1.2**    | 50.4      | 53.4       | 59.2        |
> > > >
> > > >
> > > > ### Table. 11 Computation efficiency for generating a batch of 16 images with 100 iterations on a single NVIDIA V100 GPU.
> > > >
> > > > | **Image Size**    | **Mean Time per Batch (seconds)** |
> > > > |--------------------|-----------------------------------|
> > > > | 32×32×3            | 2.06                             |
> > > > | 64×64×3            | 2.11                             |
> > > > | 84×84×3            | 2.34                             |
> > > > | 224×224×3          | 3.38                             |

---

> > > > > ### Author Response · Authors · 2025-01-23
> > > > > **Responses to the rxXC Reviewers’ Comments**
> > > > >
> > > > > $\textbf{b)}$ Additionally, the authors can use the metrics AUC-L and AUC-A to compare memory agnostic measures proposed in [1].
> > > > >
> > > > > $\textbf{Response 4(b)}$: Thank you for the thoughtful comment. Following agnostic measures proposed in [1], we have provided the performance-memory curve for CIFAR-100 in Fig. 11 and Table 8 provides the AUC values for average performance-memory (AUC-A).
> > > > >
> > > > > ### Table. 8 Memory-agnostic performance measures (AUC-A).
> > > > >
> > > > > | **Schemes**    | **CIFAR-100 (K=10)** |
> > > > > |-----------------|----------------------|
> > > > > | ASER$_\mu$     | 311.9               |
> > > > > | SCR            | 453.9               |
> > > > > | LODE           | 698.5               |
> > > > > | **Our method** | **760.3**           |
> > > > >
> > > > > $\textbf{c)}$ It can be a good contribution if authors can come up with a compute agnostic measure as the proposed method might requires additional compute to generate the images. (May be this compute is negligible in comparison to other methods, in which case the proposed algorithm would show impressive numbers.)
> > > > >
> > > > > $\textbf{Response 4(c)}$: Thank you for your thoughtful comment and suggestion. The reviewer brings up an excellent point regarding the potential need for a compute-agnostic measure to assess the method's computational requirements. Both Fig. 11 and Table 8 reveal that beyond a memory size of 44 MB (when the number of generated images exceeds $1k$), the performance of our method stabilizes. This demonstrates that our method is more scalable and extendable compared to other evaluated methods.

---

> ### Comment · Reviewer_rxXC · 2025-01-25
> **Follow Questions**
>
> I thank the authors for their detailed responses to my comments. May of my concerns and questions have been resolved. The authors have significantly improved several parts of the work.
>
> I had some follow-up questions/comments for the authors before I submit my recommendation.
> 1. [Motivation] I understand the core idea of the work and I believe the authors have made efforts to demonstrate the strengths of the work. I understand the authors intentions when they claim -
>     ( Page 2 ) "In recent works on IL, many researchers have proposed various approaches to tackle catastrophic forgetting while very few have tackled inter-task confusion explicitly. Representative methods so far could be categorized into four main groups De Lange et al. (2021): (i) replay methods, (ii) regularization-based methods, (iii) parameter isolation methods, and (iv) hybrid methods. In the replay methods, part of the data from previous tasks are stored or past data are synthesized to alleviate catastrophic forgetting Rebuffi et al. (2017); Chaudhry et al. (2019). In the regularization-based methods, regularization terms are added to the loss function, possibly combined with the approach of knowledge distillation (KD) Rannen et al. (2017). In the parameter isolation methods, the parameters of the model are controlled to address the issue of forgetting by preventing interference between the current task and past tasks Serra et al. (2018)."
>
>     - ```concern``` I have a slight hesitation to keep this part as is. I believe all the Class incremental learning algorithms would tackle both CF and ITC with different degrees. Consider the case where we give the replay base method infinite memory, you would be storing all the images seen so far and hence the final accuracy of the trained model should match the joint training approach. This implies that we were able to tackle both CF and ITC. I believe this suggests that replay-based methods can implicitly handle both ITC and CF.
>     - ```request``` I would suggest the authors reconsider making the claims milder or adding a statement **explaining that addressing CF can implicitly address ITC ( authors may choose to use the examples I stated above if it helps ) and explicitly addressing ITC can enhance the performance further.**
>
>
> 2. [Evaluation] In Response 3(a) authors state "... we reproduced the average accuracy shown in Table 1 for the methods ESMER, IMEX-Reg, CW-DPPER, and LODE, using the same data splits and task orderings. However, for the methods Gen., CF-IL, Semi, and MGI, we were unable to replicate the results based on the available source code; consequently, we extracted the results directly from the respective original papers."
>     - ```concern``` - I believe authors are yet to update table 1 and the write up of section 4.2 to reflect the newly obtained results of the implementation, describe the setup, and add citations to the repository as mentioned in this response.
>
> Kindly respond before the deadline for recommendation submission (Feb 4th, 2025).

---

> > ### Author Response · Authors · 2025-01-27
> > **Responses to the Follow Questions**
> >
> > $\textbf{Comment 5}$: I have a slight hesitation to keep this part as is. I believe all the Class incremental learning algorithms would tackle both CF and ITC with different degrees. Consider the case where we give the replay base method infinite memory, you would be storing all the images seen so far and hence the final accuracy of the trained model should match the joint training approach. This implies that we were able to tackle both CF and ITC. I believe this suggests that replay-based methods can implicitly handle both ITC and CF.
> >
> > - request: I would suggest the authors reconsider making the claims milder or adding a statement explaining that addressing CF can implicitly address ITC ( authors may choose to use the examples I stated above if it helps ) and explicitly addressing ITC can enhance the performance further.
> >
> > $\textbf{Response 5}$: Thank you for your thoughtful comment. We agree with the reviewer's suggestion to milder the paper's claims. This would provide a more accurate representation of how CIL methods handle these challenges, while still highlighting the value of explicitly addressing ITC. To address this concern, we have added following statements to the mentioned paragraph on page 2 (highlighted in blue in the updated version of the manuscript):
> >
> > “In recent works on IL, many researchers have proposed various approaches to improve the performance of class incremental learning by tackling CF. Mitigating CF can implicitly reduce ITC with different degrees, e.g., a replay base method with infinite memory should be able to address both CF and ITC. However, we argue that tackling ITC explicitly will potentially improve the performance further in realistic scenarios. Representative methods so far could be…………...”
> >
> > $\textbf{Comment 6}$: [Evaluation] In Response 3(a) authors state "... we reproduced the average accuracy shown in Table 1 for the methods ESMER, IMEX-Reg, CW-DPPER, and LODE, using the same data splits and task orderings. However, for the methods Gen., CF-IL, Semi, and MGI, we were unable to replicate the results based on the available source code; consequently, we extracted the results directly from the respective original papers."
> >
> > - concern: I believe authors are yet to update table 1 and the write up of section 4.2 to reflect the newly obtained results of the implementation, describe the setup, and add citations to the repository as mentioned in this response.
> >
> > $\textbf{Response 6}$: Thank you for your thoughtful comment. In response to this concern, we have included a detailed description of the setup, the repository utilized for the implementation of the methods, and updated Table 1. This information has been added to Section 4.2 (highlighted in blue in the updated version of the manuscript), as we intended to provide an explanation of the symbols used in Table 1 immediately following the setup for implementation.

---

> > > ### Comment · Reviewer_rxXC · 2025-02-10
> > >
> > > All of my major concerns have been addressed and I have made my recommendation to the action editor. I appreciate the author's thoughtful responses to my comments.

---

### Review · Reviewer_KcYV · 2025-01-07

**Summary Of Contributions:**

In this paper, the authors focus on the problem of class-incremental learning, where the model learns from sequential input data without accessing all past data. To this end, they aim to address the two challenges of class-incremental learning: inter-task confusion and catastrophic forgetting. In their method, they propose two regularizers in the new loss functions to combat inter-task confusion and Catastrophic Forgetting, respectively. For inter-task confusion, they propose a margin loss by using the centroids of previously learned classes as negative sample. For Catastrophic Forgetting, they use KD method in the embedding space and synthesizing past data samples by storing and re-using a past trained model. Experiments on CIFAR-10/100 and Mini-ImageNet, TinyImageNet are conducted to show the effectiveness of the proposed method.

**Audience:**

No

**Claims And Evidence:**

Yes

**Requested Changes:**

1. Parameter sensitivity analysis of $\lambda_b$. The authors should provide an analysis to show how the hyperparameter $lambda_b$ affects the data generation and the final accuracy.
2. Add a comparison of computational efficiency. See Weaknesses.
3. The authors may need to clarify the novelty of the KD regularization in the problem of class-incremental learning.

**Strengths And Weaknesses:**

Strengths:

1. The writing is basically clear. I believe readers can easily touch the core idea of this work due to the clear writing.
2. The experiments are extensive. The authors have conducted experiments on four common datasets and compared their method to many SOTA baselines, including three works published in 2024. The experimental setting is comprehensive to validate the effectiveness.

Weaknesses:

1. The KD regularization is not novel in class-Incremental learning. In the past work [1], they have proposed a KD method for the problem of class-incremental learning, so I believe this has been an old-fashion idea in this community. Although the authors use a KD method in the embedding level, but it is also not new in the area of KD. Therefore, I am not convinced by the novelty of this method due to the KD part.
2. The proposed method is complicated. In the proposed loss function, there are *five* terms with *three* hyperparameters, which are too complicated to implement in practice. Besides, the authors even add a data generation part in the method, which I believe is expensive in real world and introduce an extra hyperparameter. In my view, it would be challenging to tune the four hyperparameters in the loss function and data generation process.
3. The computational efficiency should be compared in the experiments. Due to the complexity of the proposed method, it would be better if the authors can add an experiment to compare the time-consuming of the proposed method with baselines.

[1] Kang, Minsoo, Jaeyoo Park, and Bohyung Han. "Class-incremental learning by knowledge distillation with adaptive feature consolidation." Proceedings of the IEEE/CVF conference on computer vision and pattern recognition. 2022.

---

> ### Author Response · Authors · 2025-01-23
> **Responses to the KcYV Reviewers’ Comments**
>
> We sincerely thank the reviewer for the valuable comments and suggestions. We have revised the paper based on these comments and uploaded the updated manuscript with changes highlighted in yellow.
>
> $\textbf{Comment 1}$:  Parameter sensitivity analysis of  $\lambda_b$. The authors should provide an analysis to show how the hyperparameter  $\lambda_b$ affects the data generation and the final accuracy.
>
> $\textbf{Response 1}$: Thank you for your valuable suggestion. To address this comment, we have conducted experiments on different  $\lambda_b$ and results are provided in Table 10 of section A.1.
>
> ### Table. 10 The effect of $\lambda_b$ on Average accuracy, $A_K$ (%).
>
> | **$\lambda_b$**   | **CIFAR-100 (K=10)** | **TinyImageNet (K=20)** |
> |--------------------|----------------------|-------------------------|
> | $\lambda_b = 5$    | 54.6 ± 0.3          | 53.9 ± 1.1             |
> | $\lambda_b = 10$   | **54.8 ± 0.9**      | 53.8 ± 0.9             |
> | $\lambda_b = 15$   | 54.5 ± 0.6          | **54.1 ± 1.4**         |
> | $\lambda_b = 20$   | 54.1 ± 0.7          | 53.9 ± 1.2             |
> | $\lambda_b = 50$   | 53.6 ± 0.6          | 53.3 ± 1.4             |
>
> $\textbf{Comment 2}$:  Add a comparison of computational efficiency. See Weaknesses.
>
> $\textbf{Response 2}$: Thank you for your valuable suggestion. To address this comment, we have provided consuming time for generating a batch of data in Table 11 of section A.2. Since the additional component of our method, compared to baseline approaches without data generation (e.g., fine-tuning, LwF, and EWC), is the generation of synthetic samples during training, to evaluate the computational efficiency of our approach, we measured the time required to generate a batch of 16 images across different image sizes using a single NVIDIA V100 GPU.
>
> ### Table. 11 Computation efficiency for generating a batch of 16 images with 100 iterations on a single NVIDIA V100 GPU.
>
> | **Image Size**    | **Mean Time per Batch (seconds)** |
> |--------------------|-----------------------------------|
> | 32×32×3            | 2.06                             |
> | 64×64×3            | 2.11                             |
> | 84×84×3            | 2.34                             |
> | 224×224×3          | 3.38                             |

---

> > ### Author Response · Authors · 2025-01-23
> > **Responses to the KcYV Reviewers’ Comments**
> >
> > $\textbf{Comment 3}$: The authors may need to clarify the novelty of the KD regularization in the problem of class-incremental learning.
> >
> > $\textbf{Response 3}$: Thank you for your comment regarding the contributions of our work. Below, we summarize the unique design elements and contributions of our method that distinguish it from existing research:
> >
> > 1-Novel Regularization Term for Inter-Task Confusion:
> > - We propose the Inter-Task Confusion Regularizer ($L_{TCR}$), which specifically uses the centroids of previously learned classes as negatives to minimize overlaps between classes from different tasks in the embedding space. While existing methods address catastrophic forgetting, the explicit focus on reducing inter-task confusion by leveraging embedding centroids as part of a customized loss function is a significant innovation.
> > - Unlike traditional methods, $L_{TCR}$ introduces a systematic way to maintain clear class boundaries across tasks, which has not been explicitly tackled in prior work.
> > - To the best of our knowledge, we are the first who proposed this effective mechanism to directly address the inter-task confusion.
> >
> > 2-Dual-Functionality of the Stored Model:
> > - Another novel element of this method is its use of the stored model from the previous task $M_{(k-1)}$ to generate synthetic images and serve as a teacher model for knowledge distillation.
> > - This dual-use approach is unique because most methods that rely on knowledge distillation require additional storage for data and a separate generator model (e.g., VAEs, GANs). By contrast, this paper introduces a streamlined process in which the stored model is repurposed to synthesize new data while also acting as a source of knowledge for the current task model ($M_k$).
> > - This design enables the approach to be both computationally efficient and memory-saving, which is particularly scalable to class-IL settings with extensive tasks.
> > - To the best of our knowledge, we are the first who utilize the store model for two different purposes: generating synthetic images and serving as a teacher model for knowledge distillation.
> >
> > 3-Three Novel Knowledge Distillation Losses in the Embedding Space:
> > - To address catastrophic forgetting, we introduce three KD losses that are complementing with one another:
> > 	+ MSE Loss: Aligns embeddings of the current and past models to ensure that the current model retains knowledge of past tasks.
> > 	+ Intra-Class Regularization: Preserves the embedding consistency within each class across tasks.
> > 	+ Inter-Class Regularization: Maintains the relative positions and separations between different classes in the embedding space.
> > - These losses go beyond standard KD methods by focusing on embedding-level consistency and structure preservation, which are critical for class-incremental learning.
> > - To the best of our knowledge, we are the first who proposed this set of losses in the embedding space to tackle catastrophic forgetting and inter-task confusion.
> >
> > In summary, building upon fundamental techniques such as KD and exemplar-free learning, we have introduced multiple innovative elements that significantly advance the state of the art. These contributions, combined with our practical and scalable design, set our approach apart from existing methods. We believe that the novelty and impact of our work are well-supported by the experimental results and methodological innovations presented in the paper.

---

### Review · Reviewer_jbP3 · 2025-01-10

**Summary Of Contributions:**

The authors present a method for class-incremental learning which focus on tackling two fundamental issues: the inter-task confusion and the catastrophic forgetting. The main contribution of this paper is to propose to treat the above two issues separately, i.e., the authors 1) use the centroids of previously learned classes as negative samples to design a regularization term customized to overcome inter-task confusion; 2) propose a KD method utilizing inter-class clusters, intra-class clusters and mean square distances in the embedding space to combat catastrophic forgetting. Extensive experiments performed on several classical classification tasks demonstrate that the proposed method achieves the state-of-the-art performance in the setting of class-IL.

**Audience:**

Yes

**Broader Impact Concerns:**

No major concerns.

**Claims And Evidence:**

Yes

**Requested Changes:**

- Add the results of all methods on dataset MNIST in the experiments.
- Include performance upper bound such as learning all the tasks jointly, in the experimental comparison (i.e., Table 1).
- Add in Table 3 the performance results of each method on the final task in several other cases, for example CIFAR-100 ($K=5, K=20$) and TinyImageNet ($K=5, K=10$). At the same time, rephrase the related analysis in the main text.
- Add in Table 4 the results of the evaluation of existing SOTA methods (e.g., LODE) in alleviating forgetting.
- Computation efficiency analysis.

**Typos:**
- The caption of Figure 4 is the same as that of Figure 3, which should be corrected.
- In Section 3.3 (Eq. (2)), $v\prime$ and $v’$ are not consistent. According to my understanding, $v’$ should be corrected to $v\prime$.

**Strengths And Weaknesses:**

**Strengths:**
- Overall, the paper is basically well written with clear technical contribution. The technique is well explained and is technically sound.

**Weakness:**
- MNIST is also a widely used dataset, often used alongside CIFAR-10/CIFAR-100/etc. for incremental learning algorithms validation. So it is necessary to include the results of all methods on this dataset in the experiments.
- Although Fine-tune is considered as a performance lower bound in the baselines, the experimental results do not include performance upper bound (e.g., learning all the tasks jointly), in the comparison.
- Table 3 just shows the results in two IL scenarios, and the proposed method performs best in only one case. Personally, it is inappropriate for the authors to claim that their method generally outperforms existing approaches in final task performance. Moreover, why are results only shown for the two IL scenarios? Is it because the proposed method does not perform well on other scenarios such as CIFAR-100 ($K=5, K=20$) and TinyImageNet ($K=5, K=10$)?
- In Table 4, it is not convincing to illustrate the superiority of the proposed method in alleviating forgetting by only comparing with the lower bound method Fine-tune. At the very least, a comparison should be made with existing SOTA methods, such as LODE, which performs well among the baseline methods.
- A quantitative analysis of computational efficiency is not available in the current manuscript.

---

> ### Author Response · Authors · 2025-01-23
> **Responses to the jbP3 Reviewers’ Comments**
>
> We sincerely thank the reviewer for the valuable comments and suggestions. We have revised the paper based on these comments and uploaded the updated manuscript with changes highlighted in yellow.
>
> $\textbf{Comment 1}$:  Add the results of all methods on dataset MNIST in the experiments.
>
> $\textbf{Response 1}$: Thank you for the constructive comment. To address this comment, we have added results for MNIST in Table 1.
>
> ### Table 1. Average accuracy, $A_K$ (%), after finishing the final task. Our method considerably outperforms all methods, including the coreset replay-based methods on all three datasets. The ‘MS-10,’ ‘MS-100,’ and ‘MS-Mini’ denote the memory size (MB) for CIFAR-10, CIFAR-100, and Mini-ImageNet, respectively.
>
> | **Schemes**      | **MNIST**        | **CIFAR-10**     | **CIFAR-100**    | **Mini-ImageNet** | **MS-10** | **MS-100** | **MS-Mini** |
> |-------------------|------------------|------------------|------------------|-------------------|-----------|------------|-------------|
> | Joint training    | 98.1 ± 0.06     | 83.9 ± 0.6       | 74.3 ± 1.2       | 72.1 ± 0.2        | ★         | ★          | ★           |
> | Fine-tune         | 18.2 ± 0.1      | 16.9 ± 1.1       | 5.4 ± 0.9        | 4.3 ± 0.95        | 42.6      | 42.6       | 42.6        |
> | LwF              | 20.4 ± 0.6      | 20.4 ± 0.6       | 13.5 ± 0.5       | 8.5 ± 0.50        | 42.6      | 42.6       | 42.6        |
> | EWC++            | 61.33 ± 0.10    | 18.2 ± 0.2       | 5.5 ± 0.33       | 4.3 ± 0.30        | 140.4     | 140.6      | 140.6       |
> | AGEM             | 67.9 ± 0.20     | 28.9 ± 1.20      | 14.0 ± 0.45      | 11.5 ± 0.40       | 108.9     | 108.9      | 199.4       |
> | ER               | 66.5 ± 0.35     | 49.1 ± 1.1       | 28.1 ± 1.2       | 20.9 ± 1.5        | 46.9      | 48.4       | 57.4        |
> | GSS              | 71.2 ± 0.23     | 46.4 ± 1.9       | 25.4 ± 0.55      | 20.7 ± 1.5        | 95.1      | 95.1       | 104.2       |
> | MIR              | 71.8 ± 0.30     | 49.3 ± 1.2       | 27.3 ± 1.0       | 21.8 ± 1.0        | 95.1      | 108.9      | 198.4       |
> | ASER$_\mu$       | 75.7 ± 0.40     | 50.2 ± 1.1       | 29.4 ± 0.7       | 21.0 ± 0.43       | 62.1      | 62.1       | 152.6       |
> | SCR              | 78.8 ± 0.25     | 65.6 ± 0.55      | 37.8 ± 0.6       | 35.2 ± 0.55       | 48.3      | 52.9       | 89.1        |
> | Gen.             | 93.79 ± 0.08    | 56.0 ± 0.04      | 49.5 ± 0.06      | ★                 | 99.2      | 99.2       | ★           |
> | CF-IL            | 95.3 ± 0.15     | 75.34            | ★               | ★                 | 59.2      | ★          | ★           |
> | Semi             | 92.3 ± 0.10     | 57.9 ± 1.1       | 38.9 ± 0.5       | ★                 | 96.8      | 102.3      | ★           |
> | MGI              | 90.8 ± 0.30     | 52.1 ± 2.5       | 24.1 ± 0.8       | 19.1 ± 0.9        | 96.2      | 99.8       | 135.4       |
> | ESMER            | 93.5 ± 0.20     | 73.15 ± 0.54     | 50.8 ± 0.31      | ★                 | 94.2      | 94.2       | 97.8        |
> | IMEX-Reg         | 95.7 ± 0.10     | 74.6 ± 0.18      | 50.3 ± 0.23      | ★                 | 98.9      | 98.9       | 102.5       |
> | CW-DPPER         | 91.8 ± 0.20     | 67.42 ± 1.20     | 49.15 ± 0.61     | 51.63 ± 0.11      | 99.7      | 99.7       | 135.9       |
> | LODE             | 95.5 ± 0.90     | 76.3 ± 0.90      | 51.4 ± 1.01      | 52.4 ± 0.13       | 49.7      | 63.9       | 59.7        |
> | **Our method**   | **96.1 ± 0.8**  | **76.9 ± 0.8**   | **54.8 ± 0.9**   | **54.2 ± 1.2**    | 50.4      | 53.4       | 59.2        |
>
>
> $\textbf{Comment 2}$: Include performance upper bound such as learning all the tasks jointly, in the experimental comparison (i.e., Table 1).
>
> $\textbf{Response 2}$: Thank you for the thoughtful comment. To address this comment, I added joint training as upper bound in all tables.

---

> > ### Author Response · Authors · 2025-01-23
> > **Responses to the jbP3 Reviewers’ Comments**
> >
> > $\textbf{Comment 3}$:  Add in Table 3 the performance results of each method on the final task in several other cases, for example CIFAR-100 (K=5, K=20) and TinyImageNet (K=5, K=10). At the same time, rephrase the related analysis in the main text.
> >
> > $\textbf{Response 3}$: Thank you for the thoughtful comment. To address this comment, we have added different number of tasks to Table 3 (final task accuracy) and accordingly we have rephrased the related analysis in the main text.
> >
> > **Table 3.** Final task accuracy, $A_{last}$ (%).
> >
> > | **Schemes**        | **CIFAR-100 (K=5)** | **CIFAR-100 (K=10)** | **CIFAR-100 (K=20)** | **TinyImageNet (K=5)** | **TinyImageNet (K=10)** | **TinyImageNet (K=20)** |
> > |---------------------|---------------------|----------------------|----------------------|------------------------|-------------------------|-------------------------|
> > | Joint training      | 74.32              | 74.32               | 74.32               | 66.81                 | 66.81                  | 66.81                  |
> > | SCR                | 31.22              | 28.65               | 26.10               | 29.15                 | 27.45                  | 23.11                  |
> > | ESMER              | 49.90              | 48.77               | 47.35               | 48.85                 | **47.30**              | 48.40                  |
> > | IMEX-Reg           | 50.10              | 48.54               | 46.25               | 49.65                 | 46.64                  | 41.34                  |
> > | LODE               | 51.00              | 46.31               | 46.00               | 50.20                 | 46.80                  | **45.11**              |
> > | **Our method**     | **51.25**          | **49.67**           | **47.83**           | **50.76**             | 47.13                  | 44.85                  |

---

> > > ### Author Response · Authors · 2025-01-23
> > > **Responses to the jbP3 Reviewers’ Comments**
> > >
> > > $\textbf{Comment 4}$: Add in Table 4 the results of the evaluation of existing SOTA methods (e.g., LODE) in alleviating forgetting.
> > >
> > >
> > > $\textbf{Response 4}$: Thank you for the thoughtful comment. To address this comment, we have added forgetting results of LODE in Table 4.
> > >
> > > **Table 4.** Forgetting (%) and average accuracy (%) after each task for our method, LODE, and the lower bound (i.e., fine-tuning).
> > >
> > > ### CIFAR-10
> > >
> > > | **Method**    | **Metric**    | **k=1** | **k=2** | **k=3** | **k=4** | -     | -     | -     | -     | -     |
> > > |---------------|---------------|---------|---------|---------|---------|-------|-------|-------|-------|-------|
> > > | Fine tuning   | Forgetting    | 9.5     | 12.2    | 15.7    | 28.6    | -     | -     | -     | -     | -     |
> > > |               | Accuracy      | 50.9    | 40.0    | 30.0    | 16.9    | -     | -     | -     | -     | -     |
> > > | LODE          | Forgetting    | 4.2     | 8.4     | 5.1     | 9.4     | -     | -     | -     | -     | -     |
> > > |               | Accuracy      | 90.5    | 81.9    | 79.8    | 75.4    | -     | -     | -     | -     | -     |
> > > | Our method    | Forgetting    | 4.6     | 4.7     | 5.4     | 11.1    | -     | -     | -     | -     | -     |
> > > |               | Accuracy      | 88.1    | 83.3    | 81.1    | 75.9    | -     | -     | -     | -     | -     |
> > >
> > > ---
> > >
> > > ### CIFAR-100
> > >
> > > | **Method**    | **Metric**    | **k=1** | **k=2** | **k=3** | **k=4** | **k=5** | **k=6** | **k=7** | **k=8** | **k=9** |
> > > |---------------|---------------|---------|---------|---------|---------|---------|---------|---------|---------|---------|
> > > | Fine tuning   | Forgetting    | 3.8     | 12.1    | 7.0     | 7.9     | 10.2    | 11.4    | 5.9     | 4.3     | 4.0     |
> > > |               | Accuracy      | 69.8    | 54.4    | 49.1    | 41.8    | 32.2    | 20.0    | 14.6    | 9.1     | 6.5     |
> > > | LODE          | Forgetting    | 5.0     | 10.0    | 9.2     | 2.7     | 1.5     | 1.2     | 1.2     | 1.7     | 9.4     |
> > > |               | Accuracy      | 86.3    | 74.1    | 67.3    | 64.7    | 62.8    | 61.1    | 60.4    | 58.4    | 51.5    |
> > > | Our method    | Forgetting    | 5.3     | 8.6     | 3.2     | 2.9     | 2.6     | 2.2     | 1.0     | 1.7     | 4.1     |
> > > |               | Accuracy      | 82.2    | 71.8    | 69.5    | 66.7    | 63.4    | 60.2    | 59.6    | 57.5    | 54.5    |
> > >
> > > ---
> > >
> > > ### Mini-ImageNet
> > >
> > > | **Method**    | **Metric**    | **k=1** | **k=2** | **k=3** | **k=4** | **k=5** | **k=6** | **k=7** | **k=8** | **k=9** |
> > > |---------------|---------------|---------|---------|---------|---------|---------|---------|---------|---------|---------|
> > > | Fine tuning   | Forgetting    | 12.1    | 18.9    | 14.5    | 5.3     | 9.3     | 4.8     | 5.4     | 3.5     | 2.1     |
> > > |               | Accuracy      | 51.5    | 32.1    | 20.0    | 18.9    | 11.0    | 9.8     | 6.1     | 5.5     | 5.2     |
> > > | LODE          | Forgetting    | 8.7     | 9.7     | 4.8     | 3.4     | 3.0     | 3.1     | 2.0     | 9.7     | 9.1     |
> > > |               | Accuracy      | 79.3    | 70.8    | 64.4    | 62.6    | 59.2    | 56.0    | 54.1    | 50.0    | 48.8    |
> > > | Our method    | Forgetting    | 11.1    | 6.2     | 3.4     | 3.7     | 2.2     | 2.3     | 2.0     | 4.2     | 6.6     |
> > > |               | Accuracy      | 76.8    | 71.4    | 66.8    | 64.8    | 62.3    | 59.9    | 58.0    | 56.2    | 55.3    |
> > >
> > > $\textbf{Comment 5}$: Computation efficiency analysis.
> > >
> > > $\textbf{Response 5}$: Thank you for your valuable suggestion. To address this comment, we have provided consuming time for generating a batch of data in Table 11 of section A.2. Since the additional component of our method, compared to baseline approaches without data generation (e.g., fine-tuning, LwF, and EWC), is the generation of synthetic samples during training, to evaluate the computational efficiency of our approach, we measured the time required to generate a batch of 16 images across different image sizes using a single NVIDIA V100 GPU.
> > >
> > > ### Table. 11 Computation efficiency for generating a batch of 16 images with 100 iterations on a single NVIDIA V100 GPU.
> > >
> > > | **Image Size**    | **Mean Time per Batch (seconds)** |
> > > |--------------------|-----------------------------------|
> > > | 32×32×3            | 2.06                             |
> > > | 64×64×3            | 2.11                             |
> > > | 84×84×3            | 2.34                             |
> > > | 224×224×3          | 3.38                             |
> > >
> > > $\textbf{Typos}$:
> > > - The caption of Figure 4 is the same as that of Figure 3, which should be corrected.
> > > - In Section 3.3 (Eq. (2)), $v′$ and $v^′$ are not consistent. According to my understanding, $v^′$ should be corrected to $v′$.
> > >
> > >
> > > $\textbf{Response}$: Thank you for your thoughtful feedback. We have corrected the mentioned typos.

---

> ### Comment · Reviewer_jbP3 · 2025-02-11
>
> Thank the authors for their response. After reviewing their clarifications and the revised manuscript, all my concerns have been addressed. Therefore, I recommend accepting the manuscript as is.

---

### Decision · Action_Editor_PV45 · 2025-02-11

**Recommendation:** Accept with minor revision

**Comment:**

Please include the requests from reviewers and additional experiments that have been performed.

**Audience:**

Yes researchers working on the field of continual learning.

**Claims And Evidence:**

The paper proposes a new method for incremental classification based on metric learning.

A novel loss formulation based on previous classes prototypes is proposed.
Using previous task model to generate data to be used to prevent forgetting.
A distillation loss is proposed to reduce forgetting.

Experiments are performed against various baselines showing effectiveness of the method.